# Complexity analysis of a closed-loop supply chain for power battery recycling under government subsidies

Abudureheman Kadeer⊚, Xiaobin Wang⊚⊚\*, Huanying He, Lei Dai

Xinjiang Institute of Technology, Xinjiang, China

⊚ These authors contributed equally to this work
\* 52439024@qq.com

## Abstract

Considering government subsidies for new energy electric vehicle manufacturers, the study will construct a static game model to analyze the optimal strategies of decision-makers from the perspectives of manufacturers and retailers. It will analysis the impact of key model parameters such as government subsidy criteria, the fund returns rate and market size on optimal strategy selection. Additionally, a dynamic game model will be developed to investigate the complex behavior characteristics of decision-makers, as well as the influence of different behaviors on their complex benefits. The research results show: First, government subsidy criteria, the fund return rate and market size importantly influence the optimal pricing strategies and recycling services. Second, the stability region of the equilibrium strategy is given. Moreover, when the decision-makers 's price adjustment speed exceeds a certain threshold, the equilibrium strategy undergoes complex behaviors such as bifurcation and chaos. The time-delay feedback control method is adopted to regulate the chaos. Ultimately, based on the findings, a series of targeted management insights and recommendations are proposed, indicating that while short-term government subsidies can lead to increased profits for members, excessive subsidies in the long run can result in system chaos and profit decline.

## 1. Introduction

With the evolution of the global automotive industry and the ascendancy of the carbon neutrality concept, new energy electric vehicles (EV) have gradually emerged as a pivotal choice in the quest for carbon reduction. Data from the International Energy Agency reveals that the global retired power battery capacity was merely 0.9 GWh in 2019, yet it is anticipated to surge to 120GWh by 2030 [1]. By December 2023, the number of EV in China had surpassed the milestone of 20.41 million. Predictions indicate that by 2025, the volume of retired power batteries in China will reach an

**Data availability statement:** All relevant data are within the manuscript and its Supporting Information files.

**Funding:** The author(s) received no specific funding for this work.

**Competing interests:** The authors have declared that no competing interests exist.

astounding 780,000 tons. Given the current technological landscape, power batteries have an average service life spanning approximately 5–10 years [2]. As the EV industry continues its expansion, the number of retired power batteries is poised to increase sharply.

Notably, these waste batteries contain valuable metals such as lithium and cobalt, as well as potentially harmful electrolytes, which exhibit high recycling value but also pose significant environmental risks. Therefore, the issue of power battery recycling has become a focal point for both governments and the public.

In response to global climate change and to mitigate the environmental pollution caused by retired power batteries, the European Union introduced the "Batteries and Waste Batteries Regulation" in August 2023. This regulation explicitly mandates that manufacturers be responsible for recycling all end-of-life power batteries and to utilize or dismantle them based on their actual health status. However, according to the latest data from the European Union's official website, the recycling rate of waste batteries in the EU was only 47% in 2020 [3]. To further enhance the recycling efficiency of retired power batteries, the regulation stipulates that by 2030, at least 4% of the lithium material in new batteries marketed within the EU must originate from recycled materials, with this proportion increasing to 10% by 2035 [4]. This regulation is anticipated to prompt manufacturers to explore additional channels to ensure a sufficient supply of retired power batteries. For instance, Brunp Recycling, a subsidiary of CATL, signed a tripartite agreement with Indonesia in April 2022 to jointly develop the Puqing Circular Battery Recycling Project. This project aims to preliminarily process battery waste to obtain key raw materials such as crude nickel cobalt hydroxide, manganese hydroxide, and lithium salt, which will then be transported back to China for the production of new power batteries.

The recycling of retired power batteries confronts the dual challenges of technical complexity and high costs. In response, various countries have introduced policies and regulations to promote their circular utilization. For example, the United States "Inflation Reduction Act" explicitly stipulates that retired power battery materials recycled in the United States are considered "Made in America" and are eligible for government subsidies. The European Union determines whether battery recyclers receive rewards or face penalties based on whether their recycling rates meet the standards. In 2018, China's Ministry of Industry and Information Technology issued the "Interim Measures for the Administration of the Recycling and Utilization of Power Batteries for EV" which detailed the rules for the tiered utilization of retired power batteries. Concurrently, some local governments provide subsidies, preferential loans, recycling rewards, or special funds to support the circular utilization of waste batteries. For instance, Shenzhen allocates special funds for power battery recycling and disposal at a rate of 20 yuan/kWh and provides subsidies equal to 50% of the audited amount to qualifying enterprises. Shanghai offers subsidies based on the number of batteries recycled, with 1000 yuan per set of retired power batteries [1]. Up to now, governments worldwide have introduced a series of measures, including subsidies, deposit refunds, subsidy policies, to incentivize enterprises to actively participate in the recycling of retired power batteries. However, the EV market faces significant

uncertainties and intensified competition, government subsidies criteria and their potential to promote the circular utilization of retired power batteries, as well as their impact on the supply chain, are issues that merit in-depth discussion.

The various factors outlined above collectively exert a significant influence on the decision-making process regarding the recycling of power batteries within the closed-loop supply chain (CLSC) for new energy vehicles, and their impact is not to be overlooked. Consequently, the development of an integrated the CLSC management model is of paramount importance. Because EV retailers have advantages in recycling power batteries, including reducing manufacturing costs, quickly establishing recycling channels, optimizing user experience, and promoting resource recycling, this paper constructs a "manufacturer-retailer" Stackelberg game model where retailers provide recycling services, against the backdrop of government subsidy. The "manufacturer-retailer" Stackelberg game model encapsulates both the forward supply chain of automotive products and the reverse supply chain for battery recycling. This comprehensive model takes into account the advantages of EV retailers in reducing manufacturing costs, swiftly setting up recycling networks, enhancing user satisfaction, and fostering resource circularity. By integrating these elements, the model effectively mirrors the real-world operations of the supply chain, capturing the dynamic interactions between manufacturers, retailers, and consumers in both the production and recycling processes. The main innovation of this study lies in the comprehensive model that fully considers the advantages of EV retailers in lowering manufacturing costs, rapidly establishing recycling networks, improving user satisfaction, and promoting resource circularity. By seamlessly integrating these elements, the model accurately reflects the real-world operations of the supply chain, capturing the dynamic interplay among manufacturers, retailers, and consumers throughout both the production and recycling processes. The main novelty of this work stems from the exploration of optimal strategies and complexity measurement for the CLSC of new energy electric vehicle battery recycling, considering the perspectives of manufacturers and retailers under government subsidies. Consequently, the core focus of this research is to investigate whether the application of complex system theory to THE CLSC problems can yield beneficial results. In other words, the study aims to determine whether the behavior of participants and key parameters significantly influence the outcomes of the problem, including the optimal decisions of members and the overall system performance. Furthermore, this research also aims to explore the complexity of the impact arising from such behavior. Along with answering the main research question, the following sub-questions will be answered: (1) Investigate the non-cooperative dynamic pricing strategies in a CLSC consisting of manufacturers and retailers, with retailers providing recycling services. (2) Analyze the impact of key parameters such as market potential size, The fund return rate, and government subsidies criteria on dynamic decision-making and its complexity, using complex dynamic system theory. (3) Through numerical simulations, explore the characteristics such as bifurcation, chaos, and polymorphic stability exhibited by the dynamic game system as various key parameters change. To address all these questions, this paper presents a novel modeling endeavor that incorporates the recycling service behavior of retailers into the CLCS. In the suggested model, by establishing a static Stackelberg game model and employing backward induction, we investigate the non-cooperative dynamic pricing strategies in a CLSC consisting of manufacturers and retailers, where retailers are responsible for providing recycling services. By establishing a static Stackelberg game model and utilizing the method of backward induction, our objective is to explore the non-cooperative dynamic pricing strategies within a CLSC comprising a set of manufacturers and retailers. In this framework, manufacturers entrust retailers with recycling responsibilities, while determining optimal wholesale prices, distribution prices, and recycling plans. We aim to formulate dynamic pricing strategies that maximize the overall profits for all supply chain participants. Furthermore, we analyze the influence of government subsidy criteria, fund reimbursement rate, and market size on these optimal decisions. Subsequently, based on the assumption of manufacturers' bounded rationality, a dynamic game model is established. By applying complex system theory and utilizing the Jury criterion, the stable region of the equilibrium point is determined. Then, through numerical simulation, the price adjustment rate of manufacturers is adjusted to identify the parameter ranges where bifurcation and chaos occur. The operational mechanism behind these complex phenomena is analyzed. Finally, a chaos control method is designed to mitigate and ultimately eliminate the chaotic behavior, and some management-oriented suggestions and insights are provided.

 

The results of this study indicate that market size is a crucial factor affecting the stability of the new energy vehicle supply chain. A larger market size leads to a more stable supply chain. Additionally, excessive government subsidies can have a destabilizing effect on the market. Lastly, the recycling fund reimbursement rate has a relatively minor impact on the stability of the system.

The remainder of this paper is structured in the following manner. In Sec. 2, we conduct a review of the relevant literature and highlight the key contributions presented in our paper. The model's notations and underlying assumptions are presented in Sec. 3. In Sec. 4, Initially, we formulate a single-period Stackelberg game model and derive its analytical solution through the application of backward induction. Subsequently, we develop a dynamic Stackelberg game model and assess its stability. Numerical simulations are then carried out to investigate the impact of crucial parameters on the long-term strategies and performance of channel members, as well as to explore the intricate dynamic behaviors of the CLSC system.in Sec. 5. In Sec. 6, system state variable feedback and parameter adjustment control approach method are introduced to control the system chaos. The conclusions are summarized in Sec. 7.

## 2. Literature review

This section reviews relevant literature on EV battery recycling, CLSC and government subsidies.

### 2.1 Battery recycling of EV

Many scholars both domestically and internationally are focusing on how to apply chebiometallurgy or mical methods to decompose and recycle electric vehicle power, aiming to achieve stable cycling in practical lithium-ion batteries. Currently, the recycling methods for spent LIBs primarily include hydrometallurgical, pyrometallurgical, biometallurgical processes, and direct regeneration techniques [5,6]. Pyrometallurgy entails heating the battery directly in a high-temperature smelting furnace, which results in the incineration of materials like plastics, organic matter, and graphite. The high temperatures promote the reduction of metal species, such as Co, Ni, and Cu, forming alloys with the aid of reducing agents like aluminum foil, anode graphite, and added charcoal. Meanwhile, Li, Mn, and Al are incorporated into the slag [7]. While this method allows for the large-scale processing of spent LIBs, it has limitations in recovering electrolytes, anode graphite, and valuable lithium, and it also faces challenges related to high energy consumption and the emission of harmful gases. Hydrometallurgy, on the other hand, utilizes various acids, bases, and chemicals to obtain higher-purity metals, but this often leads to the generation of acidic or alkaline wastewater, raising environmental concerns [8–11]. Biometallurgy is restricted in industrial applications due to factors like long bacterial culture cycles and increasing labor costs. The direct regeneration approach involves replenishing lithium in the spent cathode through chemical and physical methods and offers significant advantages over the aforementioned methods [10,12]. Although the direct regeneration of cathode materials is currently confined to laboratory-scale, it holds potential for large-scale industrial application in the future. In addition, some scholars have studied the issue of new energy electric vehicle battery recycling from an economic perspective. Liu et al. [13]focused on the optimization strategies for battery recycling and cascading utilization of EV from a supply chain perspective, considering factors such as channel selection, product leasing, information traceability, and supply chain collaboration. The study found that automotive manufacturers should strive to improve related technologies and select appropriate channel strategies based on market conditions. Jia et al. analyzed the impact of information asymmetry in the recycling market on the CLSC of power batteries under the extended producer responsibility system, introducing informal recycling channel competition [14]. Wei et al. established a tripartite evolutionary game model among automakers, consumers, and the government under bounded rationality. The results suggest that the government plays different roles at different stages and propose strategies for transitioning from exogenous regulation to profit-driven endogenous recycling [15]. Miao et al. analyzed fire safety issues and their causes in the field of EV, proposing management countermeasures [16]. Shen et al. discussed the impact of different government subsidy methods on the CLSC of power battery recycling, constructing models under competitive recycling scenarios and comparing results using the

Stackelberg game approach [17]. Miaomiao et al. established seven recycling models, including single, dual, and triple channels, introducing a design level parameter related to product eco-design and cascading utilization of power batteries. The study explored the impact of key parameters on design decisions, pricing, recycling rates, and profits [18]. Liu et al. constructed a dynamic game model for a CLSC consisting of battery suppliers and electric vehicle manufacturers, considering the uncertainty of residual capacity in used batteries, and analyzed the impact of battery costs and remanufacturing capacity [1]. Poschmann et al. analyzed the current status and issues of battery recycling models in automakers, proposed improvement strategies, and discussed relevant regulations and practices to inform future strategies [4]. Xing et al. introduced the current status of battery recycling in China, analyzing formal and informal recycling channels [19]. Wu et al. studied the impact of government subsidies on the CLSC, finding that subsidies to manufacturers incentivize market recovery [20]. MaLiang et al. constructed a battery recycling game model between manufacturers and retailers, exploring the effects of different recycling contracts on prices, demand, recycling rates, and profits [21]. Liujuan et al. analyzed the current status of battery recycling in China and proposed countermeasures [22]. Shen et al. compared exogenous and endogenous government subsidy scenarios, analyzing their impact on battery recycling rates [23]. Qiu et al. used evolutionary game theory to analyze the selection of recycling strategies in a secondary CLSC [24]. Chen et al. analyzed China's power battery industry policies and proposed recommendations [25].

The aforementioned literature review summarizes the technical approaches to electric vehicle battery recycling and introduces various studies, including the establishment of multiple game models. The primary focus points of these game models encompass optimization strategies, the impact of information asymmetry, fire safety issues, government subsidy methods, and the construction of different recycling models. However, a notable limitation in these studies is that they mostly conduct short-term static analysis, lacking long-term dynamic complexity analysis. This paper extends the existing research, conducting a comprehensive long-term dynamic complexity analysis that considers the evolving interactions between multiple factors such as market conditions, recycling service, government subsidy standard and supply chain dynamics over time.

## 2.2 Closed-loop supply chain

Since the concept of closed-loop supply chain was proposed by Fleischmann in [26], it has garnered increasing attention, with a growing body of literature in operations management addressing the issues pertinent to reverse logistics management for remanufacturable products. Savaskan et al. address the problem of choosing the appropriate reverse channel structure for the collection of used products from customers. The research show that simple coordination mechanisms can be designed such that the collection effort of the retailer and the supply chain profits are attained at the same level as in a centrally coordinated system [27]. The closed-loop supply chain has significant applications across various fields, which has been a focal point in recent research. On the level of healthcare supply chain management, successful applications have highlighted the significance of optimization as a focal point in recent research. Ali et al. researches the optimization of healthcare and supply chain management, revealing a multi-objective network design problem that aims to reduce costs related to temporary facilities, product transfer, and shortages in the blood supply chain. It proposes lateral freight across hospitals and introduces a robust possibilistic mixed-integer linear programming method to address distribution and locational decisions under demand uncertainty. The study employs the Torabi-Hassini methods to solve the multi-objective optimization model, revealing a total cost decrease of 10–15% [28]. Ali et al. study emphasizes the significance of optimization in healthcare and pharmaceutical supply chains, elaborating on a novel model that optimizes multiple objectives, including minimizing total costs and environmental impacts while maximizing job creation. The study addresses uncertain parameters using a fuzzy method, and evaluates various multi-objective optimization algorithms such as the multi-objective gray wolf optimizer (MOGWO), non-dominated sorting genetic algorithm II (NSGA-II), differential evolution (MODEA), and ε-constraint. It reveals that the MOGWO outperforms other approaches by generating high-quality Pareto solutions with a good spread at the boundary within a short time [29]. Saeed et al. highlights the

importance of simulation in supply chain management, illustrating how it can be used to evaluate the effects of different strategies on profit enhancement and cost reduction. It formulates a discrete event model using Arena software to assess and enhance operational efficiency in a detergent supply chain. The study elaborates on the problem, which involves multiple levels of commodities, and clarifies the overarching objective of minimizing overall costs within the system, taking into account holding costs, managing shortages, and expenses incurred due to lost sales [29]. The above three literature pieces emphasize the importance of optimization strategies and simulations in enhancing operational efficiency, reducing costs, and addressing uncertainties in healthcare and supply chain management, proposing innovative models and algorithms to achieve these objectives. In addition, there is extensive research in closed-loop supply chain management that involves the establishment of game models to analyze the impact of key parameters on optimal strategies. Ma et al. explored the impact of consumer preferences on supply chain decisions, finding that future research should focus on digital supply chains, carbon reduction policies, and supply chain competition [30]. Zheng et al. discussed the construction of reverse logistics networks for the CLSC [31]. Zhang et al. analyzed challenges facing the EV CLSC and proposed strategies [31]. Tang et al. proposed recycling and remanufacturing strategies for EV batteries [32]. Zhang et al. studied pricing and recycling in the CLSC under green preferences [33]. Gao et al. analyzed the impact of government subsidies on supply chain decisions under the EPR system [33]. Zheng et al. evaluated risks in waste electrical and electronic equipment CLSC [34]. Yin et al. studied the CLSC decisions for express packaging considering government subsidies and greenness [35]. Xing et al. analyzed the impact of consumer green preferences on the CLSC [36]. Liu et al. studied the impact of corporate social responsibility investments on the CLSC [36]. Zhang et al. explored the impact of asymmetric information on supply chain performance [37]. Niu et al. investigates the CLSC, highlighting the significance of environmental responsibility and economic benefits, with a focus on shortage channels and ordering decisions for differentiated remanufactured products [38]. Chao et al. emphasizes the importance of implementing extended producer responsibility in the recycling and reusing of household appliances, and proposes a value evaluation model to address optimal pricing problems in closed-loop supply chains, encouraging producers to fulfill their extended producer responsibility obligations [39].

Notably, there is a scarcity of literature that simultaneously examines both the forward supply chain of new energy vehicles and the reverse supply chain of battery recycling, and most articles neglect the cascaded utilization of batteries. This research aims to deepen the study by incorporating government subsidies, cascaded battery utilization, and the interplay between forward and reverse supply chains into the model, thereby contributing to a more comprehensive understanding of both the forward and reverse supply chains in the fields of new energy vehicles and battery recycling.

### 2.3 Government subsidies

Recent research has extensively explored the role of government interventions, particularly through subsidies and incentives, in promoting sustainable practices within various supply chain systems. Hao et al. highlights the importance of the extended producer responsibility system and government intervention in building sustainable economies. It develops four Stackelberg game models in a closed-loop supply chain comprising an original equipment manufacturer (OEM) and a third-party remanufacturer (TPR) under patent licensing, considering no intervention, tax subsidy, subsidy plus eco-design deduction, and deduction only modes. The impacts of different interventions on optimal production decisions, social welfare, and the environment are analyzed and compared [40]. Zi et al. investigates the optimal combination of two recycling modes and two power structures in a LC-CLSC considering the leading party's altruism and government's compound subsidy from a long-term dynamic perspective [41]. He et al. constructed an evolutionary game model among manufacturers, cascading utilization firms, and government regulators, analyzing the impact of subsidies on industrial resource recycling [42]. Qi et al. investigates a supply-chain system with technology subsidies, comprising a general contractor and two green building material manufacturers. A Stackelberg model of the green building material supply chain is developed to examine how technology subsidies influence green technology innovation among supply chain participants [43]. Fander et al studied government incentives for battery recycling. [20] analyzed recycling models based on internal and external rules. Shen et

al.[17] studied the impact of government subsidies on the CLSC decisions.[44]. Zhang et al. analyzed the impact of government subsidies and manufacturer competition on system decisions [45]. Ma et al. studied the impact of different recycling contracts on battery recycling [21]. Shi et al. studied the impact of government incentives and penalties on supply chain cooperation [46]. Ma et al. analyzed the impact of government subsidies and information asymmetry on the CLSC [47].

Notably, there is a dearth of literature that specifically examines government subsidies criteria in the context of battery recycling supply chains. Moreover, the extent and impact of government subsidies criteria on supply chain management, especially in the battery recycling sector, have remained inadequately explored. In particular, the system's complexity evolution driven by government subsidies has not been thoroughly investigated. This research intends to bridge this gap by emphasizing the analysis of the impact of government subsidies on the system's evolutionary process. Additionally, it aims to delve into the complex behavioral dynamics that arise in the absence of adequate government subsidies, ultimately contributing to a more profound understanding of the role of government subsidies in battery recycling supply chains and shedding light on the potential challenges and inefficiencies that may emerge without them.

## 3. Model description and assumptions

### 3.1 Model description

In accordance with national regulatory requirements, EV manufacturers can establish recycling channels through authorization, with BYD collaborating with its 4S retail stores for battery recycling and cascading utilization as a practical example. This chapter considers an EV supply chain comprising a manufacturer, a retailer, encompassing not only EV sales but also battery recycling and cascading utilization. The EV manufacturer produces one type of vehicle, sells it to the retailer at a wholesale price, and also sells directly to consumers, leading to sales competition within the supply chain. When an EV power battery capacity drops to 70%-80%, it must be replaced for safety and environmental reasons. Retired batteries can be repurposed or disassembled, retaining residual value. The EV manufacturer entrusts the retailer with providing battery recycling services to consumers, generating environmental and economic benefits but also increasing costs for the manufacturer. The retailer receives return of funds from the manufacturer for their recycling services. The government promotes battery recycling and cascading utilization through subsidy subsidies, providing a unit subsidy to the EV manufacturer for each battery recycled. In this Stackelberg game model, the EV manufacturer acts as the leader, setting wholesale prices and paying recycling costs, including the fund return rate to the retailer providing the service, while the EV retailer acts as a follower, purchasing EV at wholesale prices and reselling them to consumers. Consumers prefer seeking battery recycling services from the retailer, prompting manufacturers to entrust retailers with providing these services. The retailer decides on the level of battery recycling services offered to consumers, who receive a unit value corresponding to the battery's residual value after recycling.

In view of this, this paper considers the scenario where the manufacturer only entrusts the retailer to provide the recycling service for electric vehicle batteries, as depicted in the channel structure shown in Fig 1.

### 3.2 Model notation

The main symbols and their meanings involved in this chapter are shown in Table 1.

### 3.3 Model assumptions

To avoid cumbersome calculations and to maintain focus on the key aspects of the research, the corresponding model assumptions can be summarized as follows:

1. Assumes that there is only one type of EV and power battery being studied.

2. Suppose that the level of recycling service refers to the comprehensive service level of electric Power battery recycling provided by retailers. It is a comprehensive evaluation that encompasses aspects such as recycling price, distribution

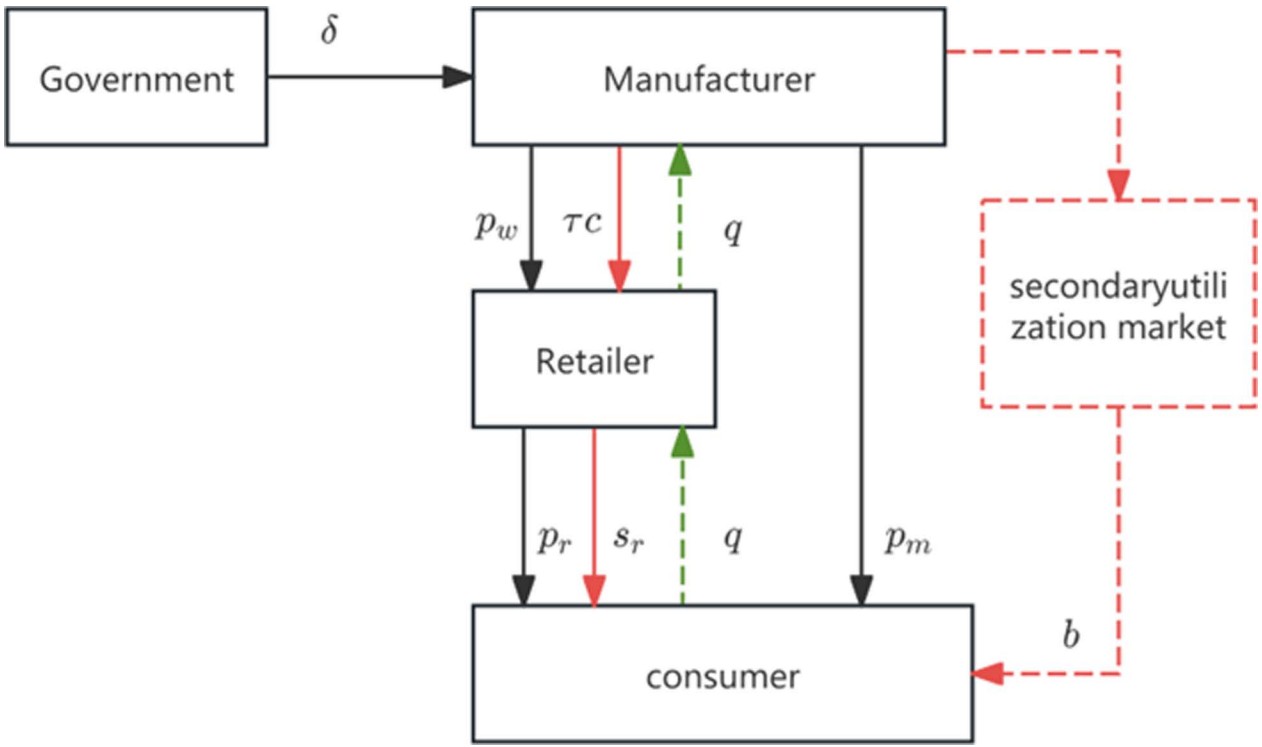

**Fig 1. Channel Diagram of Retailer Providing Recycling Services.**

of recycling outlets, recycling efficiency, service attitude, as well as technical levels in battery collection, disassembly, classification, and other aspects. Therefore, in this chapter, the recycling price of electric vehicle batteries is not considered separately.

3. Assume that the price elasticity coefficient of product sales channels is greater than the cross-price elasticity coefficient, i.e., $0 < \mu < \omega < 1$.

4. The linear additive demand function with the level recycling service adopted to characterize the special feature of market demand.

5. All the information is common knowledge for the supply chain members.

6. The cost structure of recycling service is a strictly convex function with respect to collection effort, i.e., $\frac{1}{2}\rho(s_r)^2$, which indicates the diminishing returns to investment.

## 4. Model construction and analysis

### 4.1 Static stackelberg game model

When the EV manufacturer entrusts the retailer to provide recycling services to consumers, the retailer will be responsible for providing recycling services to all consumers. In this scenario, first, the manufacturer needs to decide on the wholesale price $p_w$, direct selling price $p_m$ and pay the retailer a unit return of funds $\tau c (0 < \tau < 1)$ for the recycling of electric vehicle batteries. After purchasing EV from the manufacturer, the retailer then sets the market retail price $p_r$ for EV

Table 1. Symbols.

| Notation | Meaning |
|---|---|
| **Model parameters** | |
| $Q$ | Market demand |
| $q_m$ | Demand for direct sales channel |
| $q_r$ | Demand for retail channel |
| $q$ | The recovery volume of electric vehicle batteries by retailers |
| $\theta$ | Market share of direct selling channels |
| $a$ | Potential market size |
| $\mu$ | Consumers' cross-price sensitivity coefficient to products |
| $\gamma$ | The sensitivity coefficient to the recycling service level on channels prices |
| $\lambda$ | Recycling volume 'sensitivity coefficient to the service level recycling |
| $\omega$ | Price elasticity coefficient of consumers for products |
| $b$ | The value of retired batteries in the second-use market |
| $\rho$ | Cost coefficient of electric Power battery recycling service |
| $A$ | the number of electric vehicle batteries voluntarily returned by consumers |
| $\tau$ | The fund return rate |
| $c$ | Unit recycling cost of electric vehicle batteries paid by vehicle manufacturers |
| $q_0$ | Target recovery amount set by the government |
| $\delta$ | Correlation coefficient of government subsidy criteria |
| **Decision variables** | |
| $p_m$ | Direct selling price |
| $p_r$ | Retail price |
| $p_w$ | Wholesale price |
| $s_r$ | The level of electric Power battery recycling service provided by retailers |

and simultaneously decides on the level of recycling service $s_r$ for electric vehicle batteries provided to consumers. The decision-making sequence of the Stackelberg game model is detailed in Fig 2.

Demand function for direct sales channel:

$$q_m = \theta a - \omega p_m + \mu p_r + \gamma s \tag{1}$$

Demand function for retail channel:

$$q_r = (1 - \theta)a - \omega p_r + \mu p_m + \gamma s \tag{2}$$

Where $\omega$, $\mu$ and $\gamma$ respectively represent the consumers' price elasticity coefficient for the product, the cross-price elasticity coefficient, and the sensitivity coefficient to the recycling service level on channels prices, respectively. Once the capacity of an EV power battery decreases to 70%-80% of its original capacity, consumers can avail themselves of the retail channel to seek recycling services for the electric Power battery.

According to the findings of [13],Taking these findings into account, The recovery volume of electric vehicle batteries by retailers is:

$$q = A + \lambda s_r + b \tag{3}$$

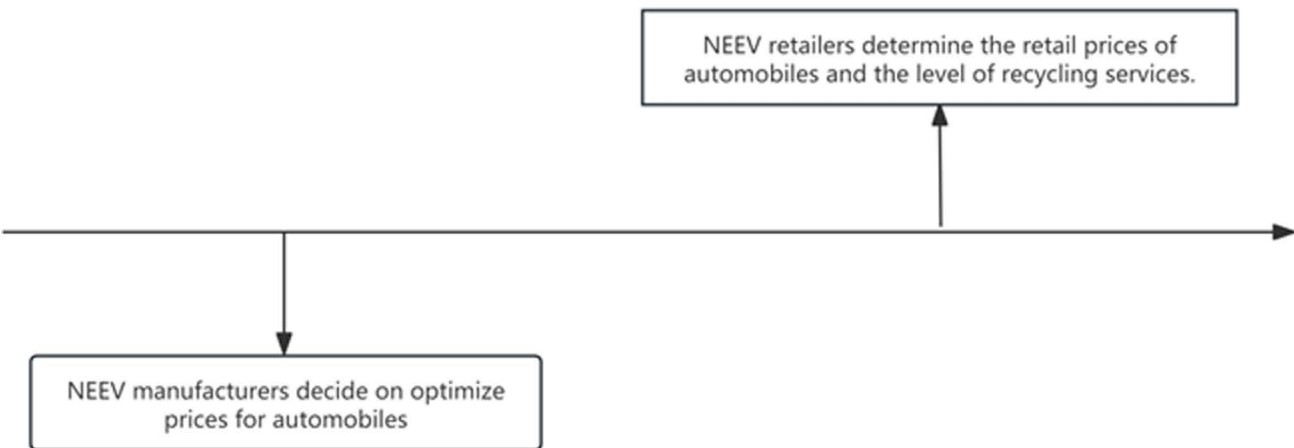

**Fig 2. The decision-making sequence of the Stackelberg game modelAssuming that the market size of EV is _a_, and the market share of the direct sales channel is** $\theta a$**, then the market share of the retail channel is** $(1-\theta)a$**.** The EV manufacturer entrusts the retailer to provide recycling services for electric vehicle batteries for all EV products sold through various channels, and the level of recycling service provided by retailers is $s_r$. The model is as follows.

Where $A$ represents the number of environmentally conscious consumers who voluntarily return electric vehicle batteries, affected by factors such as consumer environmental awareness, laws and regulations, etc. $\lambda$ represents recycling volume sensitivity coefficient to the service level recycling, the key factors influencing consumers' decision to return electric vehicle batteries include the level of recycling service, their environmental awareness, and the residual value of the battery, here, not considering the battery recycling price separately. $b$ represents the value of a unit of retired battery in the secondary use market.

Therefore, in this case, the market demand can $Q$ be expressed by the following equation.

$$Q = q_m + q_r = \theta a - \omega p_m + \mu p_r + \gamma s + (1-\theta)a - \omega p_r + \mu p_m + \gamma s = a + (\mu - \omega)(p_m + p_r) + 2\gamma s \tag{4}$$

In addition, the profit functions of the manufacturer and the retailer can be expressed as follows, adhering to academic paper standards:

$$\pi_m(p_m, p_w) = p_m q_m + p_w q_r - cq + \delta(q - q_0) \tag{5}$$

$$\pi_r(p_r, s_r) = (p_r - p_w)q_r + \tau cq - \frac{1}{2}\rho(s_r)^2 \tag{6}$$

The profit function of the EV manufacturer, as shown in Equation (5), consists of four components: the profit obtained from selling EV through the direct sales channel, the profit from wholesaling EV products to retailers, the expenditure cost for recycling electric vehicle batteries, and the recycling subsidy provided by the government. The profit function of the retailer, as shown in Equation (6), comprises three components: the profit obtained from selling EV through the retail channel, the commission for entrusted recycling provided by the manufacturer, and the expenditure cost for providing electric power battery recycling services to consumers.

**Theorem 4.1:**

When the retailer provides recycling services and the conditions

$\rho > \frac{\gamma^2(-(\mu-\omega)(\mu+3\omega))^{\frac{1}{2}}+\gamma^2\mu-\gamma^2\omega}{4(-\omega^2+\mu\omega)}$ are satisfied, the optimal pricing and optimal recycling service level for the manufacturer and retailer are respectively:

$$p_m^* = -\frac{\begin{array}{c}4a\omega^2\rho^2(\mu-\mu\theta+\omega\theta)-a\gamma^2\omega\rho(\mu-\omega-\mu\theta+3\omega\theta)\\-\gamma^3\lambda\omega(\mu+\omega)(c-\delta+c\tau)+4c\gamma\lambda\omega^2\rho\tau(\mu+\omega)\end{array}}{(\mu+\omega)\left(\gamma^4\omega+\mu\gamma^4+4\gamma^2\omega^2\rho-4\mu\gamma^2\omega\rho-8\omega^3\rho^2+8\mu\omega^2\rho^2\right)} \quad (7)$$

$$p_w^* = -\frac{\begin{array}{c}4a\omega^2\rho^2(\omega+\mu\theta-\omega\theta)\\-\gamma^3\lambda(\mu+\omega)(\delta\mu-c\mu+2c\omega-2\delta\omega+c\mu\tau+4c\omega\tau)\\-a\gamma^2\rho\left(\mu\omega-\mu^2\theta+\mu^2+2\omega^2+3\mu\omega\theta\right)+a\gamma^4(2\theta-1)(\mu+\omega)\\+4\lambda\omega\rho(\mu+\omega)(\delta\mu-c\mu+c_r\omega-\delta\omega+c\omega\tau)\end{array}}{(\mu+\omega)\left(\gamma^4\omega+\mu\gamma^4+4\gamma^2\omega^2\rho-4\mu\gamma^2\omega\rho-8\omega^3\rho^2+8\mu\omega^2\rho^2\right)} \quad (8)$$

$$p_r^* = \frac{\begin{array}{c}-a(2\theta-1)(\mu+\omega)\gamma^4+\lambda(\mu+\omega)(\delta\mu-c\mu+2c\omega-2\delta\omega+3c\omega\tau)\gamma^3\\+a\omega(2\omega+3\mu\theta-\omega\theta)\gamma^2\rho\\+2\lambda\omega(\mu+\omega)(c\mu-\delta\mu-c\omega+\delta\omega+c\mu\tau-3c\omega\tau)\gamma\rho\\-2a\omega\left(\mu^2\theta-3\omega^2\theta-\mu^2+3\omega^2+2\mu\omega\theta\right)\rho^2\end{array}}{(\mu+\omega)\left(\gamma^4\omega+\mu\gamma^4+4\gamma^2\omega^2\rho-4\mu\gamma^2\omega\rho-8\omega^3\rho^2+8\mu\omega^2\rho^2\right)} \quad (9)$$

$$s_r^* = -\frac{\begin{array}{c}a\gamma^3(\mu-\mu\theta+\omega\theta)+2\gamma^2\lambda\omega(\mu-\omega)(c-\delta+c\tau)\\+2a\gamma\omega\rho(\mu-\omega)(\theta-1)-8c\lambda\omega^2\rho\tau(\mu-\omega)\end{array}}{\gamma^4\omega+\mu\gamma^4+4\gamma^2\omega^2\rho-4\mu\gamma^2\omega\rho-8\omega^3\rho^2+8\mu\omega^2\rho^2} \quad (10)$$

**Proof:** This model belongs to a two-stage Stackelberg game model, which is solved using backward induction. In the first stage of the game, the manufacturer decides on the optimal direct sales price $p_m^*$ and optimal wholesale price $p_w^*$. In the second stage of the game, the retailer reacts by simultaneously deciding on the optimal retail price $p_r^*$ and optimal recycling service level $s_r^*$ to maximize profits.

To do this, the Hessian matrix $H_i(i=r,m)$ needs to be constructed, and the first and second partial derivatives of the retailer's profit function with respect to $p_r, s_r$ and are solved, yielding: $H_r = \begin{pmatrix} -2\omega & \gamma \\ \gamma & -\rho \end{pmatrix}$, Through calculation, the first-order leading principal minor $\left|H_r\right|_1 = -2\omega$, As well as when $\rho > \frac{\gamma^2}{2\omega}$, the second-order leading principal minor $\left|H_r\right|_2 = 2\omega\rho - \gamma^2 > 0$, then the Hessian matrix $H_r$ is negative definite. By setting the first partial derivatives $\pi_r$ with respect to $p_r$ and $s_r$ in the expression given by (6) to zero, we can obtain the retailer's reaction function (RF) to the manufacturers through backward induction, as follows:

$$p_r^{RF} = \frac{-p_w\gamma^2 + c\lambda\tau\gamma + \alpha\rho - \alpha\rho\theta + \mu\rho p_m + \omega\rho p_w}{2\omega\rho - \gamma^2} \quad (11)$$

$$s_r^{RF} = \frac{\alpha\gamma - \alpha\gamma\theta + \gamma\mu p_m - \gamma\omega p_w + 2c\lambda\omega\tau}{2\omega\rho - \gamma^2} \quad (12)$$

Substituting (11) and (12) into (5) yields the manufacturer's profit function based on the retailer's reaction function.

In the first stage of the game, the manufacturer needs to decide on the optimal direct sales price $p_m^*$ and optimal wholesale price $p_w^*$ to obtain optimal profits. By solving the first and second partial derivatives $\pi_m$ with respect to $p_m$ and $p_w$ in equation (5), we obtain the Hessian matrix: $H_m = \begin{pmatrix} \frac{2\gamma^2\mu + 2\gamma^2\omega + 2\rho\mu^2 - 4\rho\omega^2}{2\omega\rho - \gamma^2} & \frac{-(\gamma^2\mu + \gamma^2\omega - 2\mu\omega\rho)}{2\omega\rho - \gamma^2} \\ \frac{-(\gamma^2\mu + \gamma^2\omega - 2\mu\omega\rho)}{2\omega\rho - \gamma^2} & \frac{2\omega^2\rho}{2\omega\rho - \gamma^2} \end{pmatrix}$,.

Through calculation, it is obtained that when $\rho > \frac{\gamma^2\mu + \gamma^2\omega}{\omega^2 - \mu^2}$ for $\left|H_m\right|_1 < 0$, and if $\left|H_m\right|_2 > 0$, $\rho$ satisfies the inequality:
$\gamma^4\omega + \mu\gamma^4 + 4\gamma^2\omega^2\rho - 4\mu\gamma^2\omega\rho - 8\omega^3\rho^2 + 8\mu\omega^2\rho^2 < 0$,

i.e., $\rho > \frac{\gamma^2(-(\mu - \omega)(\mu + 3\omega))^{\frac{1}{2}} + \gamma^2\mu - \gamma^2\omega}{4(-\omega^2 + \mu\omega)}$ or $\rho < -\frac{\gamma^2(-(\mu - \omega)(\mu + 3\omega))^{\frac{1}{2}} + \gamma^2\mu - \gamma^2\omega}{4(-\omega^2 + \mu\omega)}$

(This option is discarded and does not conform to reality, because of p > 0).

Then if $\left|H_m\right| = -\frac{(\mu + \omega)\left(\gamma^4\omega + \mu\gamma^4 + 4\gamma^2\omega^2\rho - 4\mu\gamma^2\omega\rho - 8\omega^3\rho^2 + 8\mu\omega^2\rho^2\right)}{(2\omega\rho - \gamma^2)^2} > 0$, the hessian matrix is negative definite, indicating that the manufacturer's profit function is a concave function with respect to $\pi_m$. Therefore, there exists a unique optimal solution with respect to $p_m$ and $p_w$. By solving the first partial derivatives $\pi_m$ with respect to $p_m$, $p_w$ and setting them equal to zero, we obtain:

$$p_m^* = -\frac{\begin{array}{c} 4a\omega^2\rho^2(\mu - \mu\theta + \omega\theta) - a\gamma^2\omega\rho(\mu - \omega - \mu\theta + 3\omega\theta) \\ -\gamma^3\lambda\omega(\mu + \omega)(c - \delta + c\tau) + 4c\gamma\lambda\omega^2\rho\tau(\mu + \omega) \end{array}}{(\mu + \omega)\left(\gamma^4\omega + \mu\gamma^4 + 4\gamma^2\omega^2\rho - 4\mu\gamma^2\omega\rho - 8\omega^3\rho^2 + 8\mu\omega^2\rho^2\right)}$$

$$p_w^* = -\frac{\begin{array}{c} 4a\omega^2\rho^2(\omega + \mu\theta - \omega\theta) \\ -\gamma^3\lambda(\mu + \omega)(\delta\mu - c\mu + 2c\omega - 2\delta\omega + c\mu\tau + 4c\omega\tau) \\ -a\gamma^2\rho\left(\mu\omega - \mu^2\theta + \mu^2 + 2\omega^2 + 3\mu\omega\theta\right) + a\gamma^4(2\theta - 1)(\mu + \omega) \\ +4\gamma\lambda\omega\rho(\mu + \omega)(\delta\mu - c\mu + c\omega - \delta\omega + c\omega\tau) \end{array}}{(\mu + \omega)\left(\gamma^4\omega + \mu\gamma^4 + 4\gamma^2\omega^2\rho - 4\mu\gamma^2\omega\rho - 8\omega^3\rho^2 + 8\mu\omega^2\rho^2\right)}$$

Substituting $p_m^*, p_w^*$ into (11) and (12), we obtain:

$$p_r^* = \frac{\begin{array}{c} -a(2\theta - 1)(\mu + \omega)\gamma^4 + \lambda(\mu + \omega)(\delta\mu - c\mu + 2c\omega - 2\delta\omega + 3c\omega\tau)\gamma^3 \\ +a\omega(2\omega + 3\mu\theta - \omega\theta)\gamma^2\rho \\ +2\lambda\omega(\mu + \omega)(c\mu - \delta\mu - c\omega + \delta\omega + c\mu\tau - 3c\omega\tau)\gamma\rho \\ -2a\omega\left(\mu^2\theta - 3\omega^2\theta - \mu^2 + 3\omega^2 + 2\mu\omega\right)\rho^2 \end{array}}{(\mu + \omega)\left(\gamma^4\omega + \mu\gamma^4 + 4\gamma^2\omega^2\rho - 4\mu\gamma^2\omega\rho - 8\omega^3\rho^2 + 8\mu\omega^2\rho^2\right)}$$

$$s_r^* = -\frac{\begin{array}{c} a\gamma^3(\mu - \mu\theta + \omega\theta) + 2\gamma^2\lambda\omega(\mu - \omega)(c - \delta + c\tau) \\ +2a\gamma\omega\rho(\mu - \omega)(\theta - 1) - 8c\lambda\omega^2\rho\tau(\mu - \omega) \end{array}}{\gamma^4\omega + \mu\gamma^4 + 4\gamma^2\omega^2\rho - 4\mu\gamma^2\omega\rho - 8\omega^3\rho^2 + 8\mu\omega^2\rho^2}$$

Theorem 4.1 outlines the necessary constraints to guarantee the existence of an equilibrium solution in the static game model. The above constraints serve as a guide for decision-makers, offering them a range of effective parameters to sustain the game system in an equilibrium state. Additionally, supply chain participants are motivated to adjust their parameters in alignment with these constraints, in pursuit of maximizing their profits.

Further, based on the aforementioned optimal strategies, the optimal profits for the manufacturer and the retailer are derived and denoted as $\pi_m^*$ and $\pi_r^*$, representing the optimal profits of the manufacturer and the retailer respectively.

In the subsequent analysis, we will concentrate on investigating the influence of certain pivotal parameters on the optimal strategy adopted by decision makers and the performance of supply chain members. Propositions is articulated as follows:

**Theorem 4.2:**

$$\frac{\partial p_m^*}{\partial \delta} < 0, \frac{\partial p_m^*}{\partial \tau} < 0, \frac{\partial p_m^*}{\partial a} < 0$$

$$\frac{\partial p_w^*}{\partial \delta} < 0, \frac{\partial p_w^*}{\partial \tau} < 0, \frac{\partial p_w^*}{\partial a} < 0$$

$$\frac{\partial p_r^*}{\partial \delta} < 0, \frac{\partial p_r^*}{\partial \tau} > 0, \frac{\partial p_r^*}{\partial a} < 0$$

$$\frac{\partial s_r^*}{\partial \delta} > 0, \frac{\partial s_r^*}{\partial \tau} > 0, \frac{\partial s_r^*}{\partial a} < 0$$

**Proof:** The validity of this theorem is established in [48], and thus its proof is omitted for brevity.

Theorem 4.2 indicates

(*i*) Government subsidy criteria exerts a negative effect direct sales channel prices $p_m$ and a positive impact on retail channel prices (wholesale price $p_w$ and retail price $p_r$). which means the higher degree government subsidy criteria, the higher the retail channel price and lower prices in the direct sales channel. In short, government subsidy policies in this context serve as a regulatory mechanism to adjust the price disparities observed across different sales channels, aiming to promote sales in the retail channel while suppressing price increases in the direct sales channel. Additionally, it has a positive effect on the level of electric Power battery recycling service provided by retailers. It is straightforward that the subsidy policy reduces retailers' operating costs, increases investment returns, enhances market competitiveness, guides consumer behavior, internalizes environmental externalities, and exerts policy guidance and regulatory promotion effects, so the level of recycling service charged by it will be raised.

(*ii*) The fund return rate of manufacturer has a negative effect on manufacturers' price (direct sales price $p_m$ and wholesale price $p_w$) and a positive effect on retailers' prices. The reason why is that the fund return rate promotes retailers to offset costs and increase prices while maintaining competitiveness, or manufacturers can adjust their wholesale prices in response to the rebate, subsequently influencing retail pricing.

(*iii*) The higher potential market size is, the lower channel prices are. In larger markets, there is increased competition among suppliers, including manufacturers and retailers, to capture a larger share of the market. This competition often drives down prices as suppliers seek to attract customers and gain a competitive edge. Additionally, in larger markets, suppliers may benefit from economies of scale, allowing them to produce and distribute goods more efficiently, which can also contribute to lower prices. Therefore, the combination of increased competition and potential for cost savings in larger markets results in lower channel prices.

**Theorem 4.3:**

$$\frac{\partial \pi_m^*}{\partial \delta} > 0, \frac{\partial \pi_m^*}{\partial \tau} > 0, \frac{\partial \pi_m^*}{\partial a} > 0$$

$$\frac{\partial \pi_r^*}{\partial \delta} > 0, \frac{\partial \pi_r^*}{\partial \tau} > 0, \frac{\partial \pi_r^*}{\partial a} > 0$$

**Proof:** The theorem's validity has been confirmed in [48], so its proof is not included here for the sake of conciseness.

Theorem 4.3 indicates

(*i*) The enhancement of government subsidy criteria exerts a dual impact on manufacturer and retailer profits. For manufacturers, while subsidies may lead to a decrease in direct sales prices, they can offset this by increasing sales volumes, particularly in the retail channel, thereby potentially boosting overall profits. Subsidies also reduce production costs and encourage higher investment returns, indirectly benefiting manufacturers. For retailers, government subsidies directly elevate retail channel prices and lower operating costs, resulting in increased profit margins. Additionally, subsidies enhance market competitiveness, guide consumer behavior towards more environmentally friendly options, and internalize externalities, all contributing to higher retailer profits.

(*ii*) The fund return rate positively influences retailer profits by enabling cost recovery and price adjustments that maintain competitiveness. As manufacturers receive faster returns on their investments, they may adjust wholesale prices to accommodate retailers, indirectly affecting retail pricing and profit margins. Retailers, in turn, can leverage these adjustments to increase their prices while remaining attractive to consumers. This dynamic allows both manufacturers and retailers to potentially increase profits through more efficient capital utilization and pricing strategies.

(*iii*) The expansion of market size typically leads to increased competition among suppliers, including manufacturers and retailers. While this competition often drives down prices, it also opens up new sales opportunities and potential for profit growth. In larger markets, suppliers can benefit from economies of scale, reducing production and distribution costs and enabling them to maintain profitability even at lower prices. Therefore, despite the price competition, the overall increase in sales volume and cost efficiency can lead to higher profits for both manufacturers and retailers as the market size grows.

## 4.2 Dynamic Stackelberg game model

A dynamic Stackelberg game model has been established. Given that the price decision-making is constrained by objective conditions such as decision-making capabilities, decision-makers cannot obtain complete market information. Assuming that the manufacturer is boundedly rational, as described in [47,48], decisions in the next period should undergo bounded rational adjustments, which means that price decisions are based on a partial estimation of the marginal profit in the current period. If the marginal profit in period t is positive, the manufacturer will increase prices in period $t + 1$. Otherwise, the manufacturer will decrease them. The model can be constructed as follows:

$$
\begin{cases}
p_w(t+1) = p_w(t) + \alpha_1 p_w(t)\frac{\partial \pi_m(t)}{\partial p_w(t)} \\
p_m(t+1) = p_m(t) + \alpha_2 p_m(t)\frac{\partial \pi_m(t)}{\partial p_m(t)}
\end{cases}
$$

Where $\alpha_i$ represents the manufacturer's price adjustment speed, and $\frac{\partial \pi_m(t)}{\partial p_w(t)}, \frac{\partial \pi_m(t)}{\partial p_m(t)}$ respectively represent the marginal profits, which are obtained from the equations (11) (12) (6). we obtain the discrete dynamic model related to the manufacturer as follows:

$$
\begin{cases}
p_w(t+1) = p_w(t) + \alpha_1 p_w(t)\left(\dfrac{\gamma^2(\alpha - \alpha\theta + \mu p_m(t) - \omega p_w(t)) + 2c\lambda\omega\tau}{2\omega\rho - \gamma^2} \right. \\
\qquad \left. + \mu p_m(t) - \dfrac{\omega(-p_w(t)\gamma^2 + c\lambda\tau\gamma + \alpha\rho - \alpha\rho\theta + \mu\rho p_m(t) + \omega\rho p_w(t))}{2\omega\rho - \gamma^2} - \alpha(\theta - 1)\right) \\
p_m(t+1) = p_m(t) + \alpha_2 p_m(t)\left(\alpha\theta + \dfrac{\alpha\gamma^2 - \alpha\gamma^2\theta + \gamma^2\mu p_m(t) - \gamma^2\omega p_w(t) + 2c\gamma\lambda\omega\tau}{2\omega\rho - \gamma^2} \right. \\
\qquad \left. + \dfrac{\mu(-p_w(t)\gamma^2 + c\lambda\tau\gamma + \alpha\rho - \alpha\rho\theta + \mu\rho p_m(t) + \omega\rho p_w(t))}{2\omega\rho - \gamma^2} + \mu p_w(t) - 2\omega p_m(t)\right)
\end{cases}
$$

(13)

Given that the retailer's decisions are dependent on the manufacturer's decisions, we derive the retailer's discrete dynamic model as follows:

$$\begin{cases} p_r(t) = \frac{-p_w(t)\gamma^2 + c\lambda\tau\gamma + \alpha\rho - \alpha\rho\theta + \mu\rho p_m(t) + \omega\rho p_w(t)}{2\omega\rho - \gamma^2} \\ s_r(t) = \frac{\alpha\gamma - \alpha\gamma\theta + \gamma\mu p_m(t) - \gamma\omega p_w(t) + 2c\lambda\omega\tau}{2\omega\rho - \gamma^2} \end{cases} \tag{14}$$

The final result is the following dynamic discrete game system model:

$$\begin{cases} p_m(t+1) = p_m(t) + \alpha_2 p_m(t)(\alpha\theta + \gamma\dfrac{\alpha\gamma - \alpha\gamma\theta + \gamma\mu p_m(t) - \gamma\omega p_w(t) + 2c\lambda\omega\tau}{2\omega\rho - \gamma^2} \\ \quad + \dfrac{\mu(-p_w(t)\gamma^2 + c\lambda\tau\gamma + \alpha\rho - \alpha\rho\theta + \mu\rho p_m(t) + \omega\rho p_w(t))}{2\omega\rho - \gamma^2} + \mu p_w(t) - 2\omega p_m(t)) \\ p_w(t+1) = p_w(t) + \alpha_1 p_w(t)(\dfrac{\gamma^2(\alpha - \alpha\theta + \mu p_m(t) - \omega p_w(t)) + 2c\lambda\omega\tau}{2\omega\rho - \gamma^2} \\ \quad + \mu p_m(t) - \dfrac{\omega(-p_w(t)\gamma^2 + c\lambda\tau\gamma + \alpha\rho - \alpha\rho\theta + \mu\rho p_m(t) + \omega\rho p_w(t))}{2\omega\rho - \gamma^2} - \alpha(\theta - 1)) \\ p_r(t) = \frac{-p_w(t)\gamma^2 + c\lambda\tau\gamma + \alpha\rho - \alpha\rho\theta + \mu\rho p_m(t) + \omega\rho p_w(t)}{2\omega\rho - \gamma^2} \\ s_r(t) = \frac{\alpha\gamma - \alpha\gamma\theta + \gamma\mu p_m(t) - \gamma\omega p_w(t) + 2c\lambda\omega\tau}{2\omega\rho - \gamma^2} \end{cases} \tag{15}$$

## 4.3 Model analysis

**4.3.1 Equilibrium point.** The discrete dynamic system (15) can be analyzed through $p_w(t+1) = p_w(t)$, $p_m(t+1) = p_m(t)$ to obtain four equilibrium points, the specific description is as follows: $E_1\ (0,0)$, $E_2(\frac{\omega(a\rho + c\gamma - \delta\gamma - a\rho\theta + c\gamma\tau)}{\gamma^2\mu + \gamma^2\omega - 2\mu\omega\rho}, 0)$, $E_3\left(0, \frac{\omega(a\gamma^2 + a\mu\rho - 2a\gamma^2\theta - c\gamma\lambda\mu + \delta\gamma\lambda\mu - a\mu\rho\theta + 2a\omega\rho\theta + c\gamma\lambda\mu\tau + 2c\gamma\lambda\omega\tau)}{\gamma^2\mu + \gamma^2\omega - 2\mu\omega\rho}\right)$, $E_4(p_m^*, p_w^*)$.

Obviously, upon examination, it is evident that among the four equilibrium solutions presented, solely $E_4$ possesses non-zero values across all its components, whereas the remaining three solutions, $E_1, E_2$ and $E_3$, represent boundary cases. In economic contexts, decision-makers typically do not permit decision variables to be zero, implying that $E_4$ is the only stable equilibrium among the four. This stability assertion can be further validated through the computation of the Jacobian matrix for each equilibrium solution.

**4.3.2 Stability analysis of equilibrium solutions.** To analyze the stability of equilibrium solutions $E_4$ in the Stackelberg model, the Jacobian matrix of the discrete game system is calculated and presented as follows:

$$J = \begin{pmatrix} \frac{\alpha_1 F_1}{2\omega\rho - \gamma^2} + \frac{\alpha_1 p_m F_2}{2\omega\rho - \gamma^2} + 1 & \frac{-\alpha_1 p_m F_3}{2\omega\rho - \gamma^2} \\ \frac{-\alpha_2 p_m F_4}{2\omega\rho - \gamma^2} & -\frac{\gamma^2 - 2\omega\rho + \alpha_2 F_5 + \alpha_2\gamma^2 F_6}{2\omega\rho - \gamma^2} \end{pmatrix}$$

where,

$$F_1 = \alpha\gamma^2 + 2\gamma^2\mu p_m - \gamma^2\mu p_w + 2\gamma^2\omega p_m - \gamma^2\omega p_w$$

$$+ 2\mu^2\rho p_m - 4\omega^2\rho p_m + \alpha\mu\rho - 2\alpha\gamma^2\theta - c\gamma\lambda\mu + \delta\gamma\lambda\mu$$

$$- \alpha\mu\rho\theta + 2\alpha\omega\rho\theta + 2\mu\omega\rho p_w + c\gamma\lambda\mu\tau + 2c\gamma\lambda\omega\tau$$

$$F_2 = 2\gamma^2\mu + 2\gamma^2\omega + 2\rho\mu^2 - 4\rho\omega^2$$

$$F_3 = \gamma^2\rho + \gamma^2\omega - 2\mu\omega\rho$$

$$F_4 = \gamma^2\mu + \gamma^2\omega - 2\mu\omega$$

$$F_5 = \gamma^2\mu p_m - c\gamma\lambda\omega - \alpha\omega\rho + \delta\gamma\lambda\omega + \alpha\omega\rho\theta - 2\mu\omega\rho p_m - c\gamma\lambda\omega\tau$$

$$F_6 = p_m + 4\alpha_2\omega^2\rho p_w$$

The stability of the equilibrium points will be determined by the properties of the eigenvalues of the Jacobian matrix calculated at the equilibrium points. Then, we will substitute the values of $E_1, E_2$ and $E_3$ into J respectively and present the following propositions.

**Theorem 4.3**: All boundary equilibrium points, namely $E_1, E_2$ and $E_3$ are unstable points.

**Proof**: The proof method for this theorem, as cited in the references, involves [30,48,49]

To investigate the stability of the equilibrium on point $E_4$, we obtained the Jacobian matrix regarding $J(E_4)$, which is as follows:

$$J(E_4) = \begin{pmatrix} \frac{\alpha_l F_1(p_m^*, p_w^*)}{2\omega\rho - \gamma^2} + \frac{\alpha_l p_m^* F_2}{2\omega\rho - \gamma^2} + 1 & \frac{-\alpha_1 p_m^* F_3}{2\omega\rho - \gamma^2} \\ \frac{-\alpha_2 p_m^* F_4}{2\omega\rho - \gamma^2} & -\frac{{}^2 - 2\omega\rho + \alpha_2 F_5(p_m^*) + \alpha_2{}^2 F_6(p_m^*, p_w^*)}{2\omega\rho - 2} \end{pmatrix}$$

The corresponding characteristic equation is: $r^2 - Tr(J)r + Det(J) = 0$

where $Tr(J)$ is the trace and Det(J) is the determinant of the Jacobian matrix.

When $(Tr(J))^2 - 4Det(J) > 0$, it indicates that the characteristic values of the characteristic polynomial of the Nash equilibrium solution are real numbers.

According to the Jury criterion, a sufficient and necessary condition for the local stability of the Nash equilibrium solution is that all eigenvalues of $J(E_4)$ lie within the unit circle in the complex plane. Therefore, the following three conditions should be simultaneously satisfied, as mentioned above:

$$\begin{cases} (i) : 1 + Tr(J) + Det(J) > 0 \\ (ii) : 1 - Tr(J) + Det(J) > 0 \\ (iii) : Det(J) - 1 < 0 \end{cases}$$

(16)

Therefore, by carefully adjusting the parameters $\alpha_1$ and $\alpha_2$, we can define and optimize the stability interval of the Nash equilibrium point. This adjustment process is crucial because if any one of the above three stability conditions is not satisfied while the other two conditions are still maintained, the system may encounter different types of dynamic bifurcation phenomena.

It is worth noting that although changes in $\alpha_1$ and $\alpha_2$ profoundly affect the stability patterns of the system, they do not directly influence the numerical position of the Nash equilibrium point itself. The economic interpretation of this stable region is that it defines a range within which, as long as the $\alpha_i$ parameters fall, regardless of the initial strategies or values adopted by the manufacturers, the system will ultimately converge to the Nash equilibrium point after a finite number of market games, achieving stability and predictability in the market state. This process not only enhances the resilience of market mechanisms but also provides valuable reference for policymakers to promote the healthy and orderly development of the market through fine-tuning parameters.

# 5. Numerical simulation

Based on the constraint (16) and drawing on literature [21,30,47,50–52] experiences, We set the parameters $\beta = 3.65; \rho = 2; \omega = 0.4; \gamma = 0.6; \theta = 0.4; \mu = 0.2; \lambda = 0.2; c = 6$ and the initial values $p_w = 10, p_m = 15$ in the system (17).

## 5.1 Stability analysis of equilibrium solutions

In this chapter, we consider the stability analysis of the system with respect to $\delta, \tau$ and $a$, the stability of the equilibrium $E_4$ is demonstrated in Fig 3. As can be seen from Fig 3, the stable regions of the Nash equilibrium point with respect to $\alpha_1$ and $\alpha_2$ under different parameter values are given, respectively.

In Fig 3(a), we fix $\tau = 0.2$ $a = 10$, the red stable region is presented when $\delta$ = 2, the blue stable region is shown when $\delta = 2$, the blue stable region is shown when $\delta = 12$, and the purple stable region is displayed when $\delta = 22$, One knows that the scope of $a_1$ are decreased as $\delta$ increases, while $a_2$ remains almost unaffected. the size of stability region of the system is reduced. Similarly, in Fig 3(b), $\delta = 2$ and $a = 10$ are fixed. We assign $\tau = 0.2, 0.5$ and $0.9$ and the stable regions are shown in red, blue and yellow, respectively. We see that as $\tau$ increases, scope of $\alpha_1$ is increases, the both scopes of $\alpha_1$ and $\alpha_2$ is increases and the size of stability region of the system is reduced, however, the stability region is insensitive to the parameter $\tau$. Fig 3(c) shows the stable regions with respect to $a$ =10, 12 and 14 when $\delta = 2$, $\tau = 0.2$ are fixed, which correspond to the region enclosed by yellow, blue and red regions respectively. We see the scopes of $\alpha_1$ and $a_2$ is increased as $a$ increases, and the size of stability region of the system is reduced. In addition, it is obvious that the stability region is more sensitive to $\alpha_1$ and $a_2$, resulting in significant changes in the stable region.

From the above analysis, we conclude that Consumers' subjective attitude, commissioned recycling cost coefficient and government subsidy have a destabilization effect on the Nash equilibrium point. In addition, under certain conditions, government subsidies can inhibit the stability of the supply chain, the reason lies in that excessive or improperly structured subsidies may disrupt the market balance and incentivize behaviors that undermine the stability of the supply chain. Alternatively, government subsidies, if not well-designed, can create distortions that destabilize the supply chain dynamics. Additionally, since the Commissioned recycling cost coefficient has little influence on the stability region, the manufacturer can determine the recycling rebate rate for retailers without considering the stability region, as long as maximum profit is ensured. Furthermore, as the market size has a significant and positive impact on the stability region of the equilibrium point, it requires manufacturer to accelerate the speed of direct sales price and wholesale price. during market booms and to decelerate the speed of direct sales price and wholesale price adjustments during market downturns.

## 5.2. Local bifurcation analysis

Fig 4 illustrates the behavior of the decision variables with respect to the adjustment speed $\alpha_1$, maintaining $\alpha_2$ at 0.05. By examining Fig 4(a)–4(c), it is evident that a low adjustment speed for $\alpha_1$ (specifically, $\alpha_1 < 0.3326$) ensures system stability. When $\alpha_1$ reaches 0.3326, the initial bifurcation point emerges. As $\alpha_1$ is further increased, the Nash equilibrium point experiences period-doubling (or flip) bifurcations, ultimately transitioning into a chaotic state. The Largest Lyapunov Exponent (*LLE*) diagram, depicted in Fig 4(d), confirms the chaotic behavior exhibited by system (16). Specifically, when *LLE*<0, it indicates a stable system state; an *LLE*=0 corresponds to the bifurcation point; and when *LLE*>0, it signifies the onset of complex dynamic behavior.

Fig 5 depicts the behavior of the decision variables with respect to the adjustment speed $\alpha_2$, while keeping $\alpha_1$ fixed at 0.17. Upon analyzing Fig 5(a) through 5(c), it becomes apparent that a low adjustment speed for $\alpha_2$ (precisely, $\alpha_2 < 0.2326$) maintains the stability of the system. When $\alpha_2$ reaches the value of 0.2326, the first bifurcation point manifests. As $\alpha_2$ continues to increase, the Nash equilibrium point undergoes period-doubling (or flip) bifurcations, eventually leading to a chaotic state. The diagram of the *LLE*, presented in Fig 5(d), corroborates the chaotic behavior demonstrated by system (16). Particularly, when *LLE*<0, it signifies a stable system state; when *LLE*=0, it corresponds to the bifurcation point; and when *LLE*>0, it denotes the emergence of complex dynamic behavior.

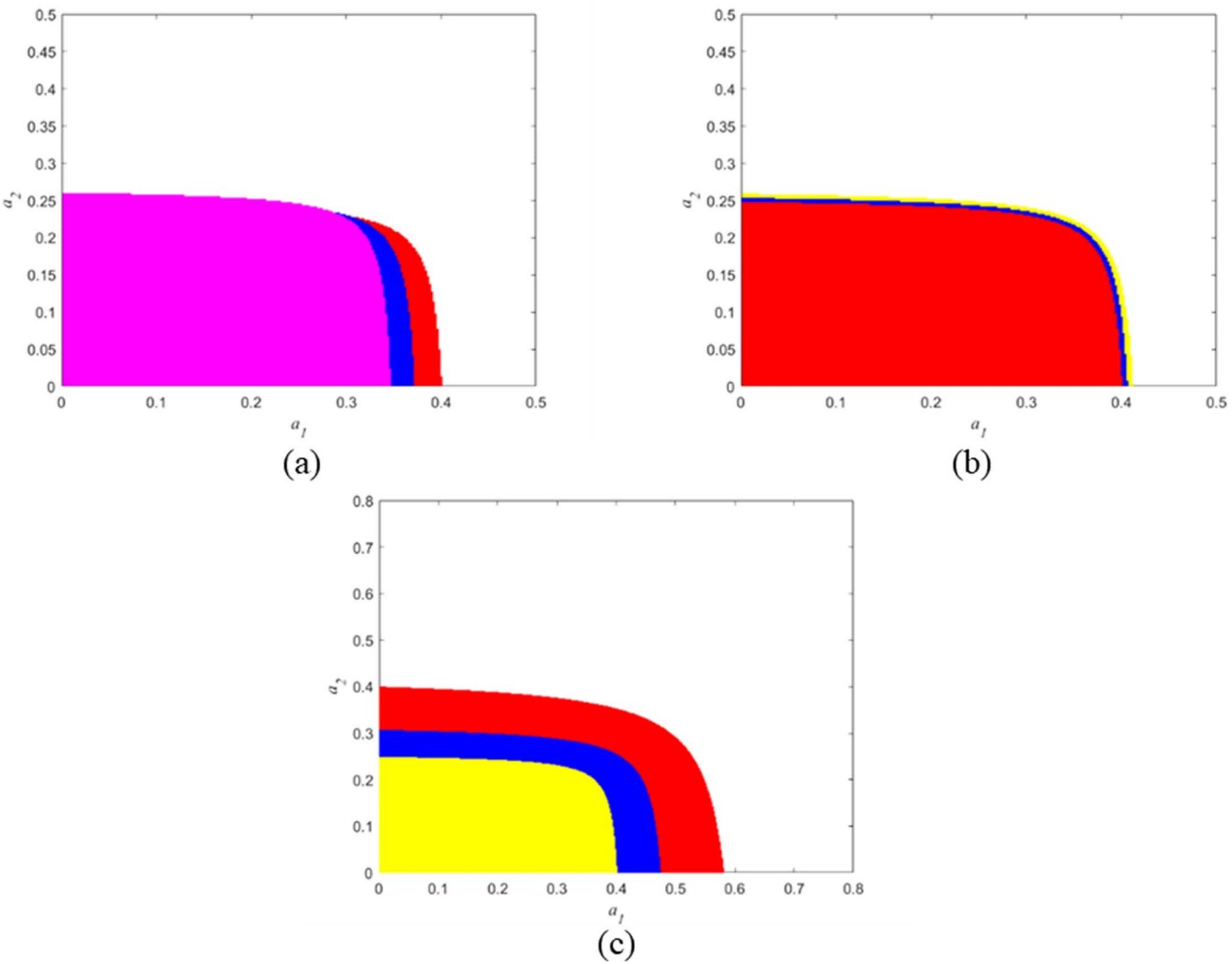

**Figs 3. The influence of parameters $\delta, \tau$ and $a$ on the stable regions.** (a) $a = 10$ , $\tau = 0.2$, (b) $\delta = 2$, $a = 10$ and (c) $\delta = 2$, $\tau = 0.2$.

## 6. Chaos control

After performing the aforementioned numerical simulations, it has been observed that the rate of parameter adjustment significantly impacts the stability of the dynamic CLSC system (16). When the system deviates from the stability zone and exhibits more intricate behaviors, the overall efficiency of the CLSC system diminishes considerably. In most instances, the emergence of chaotic behavior is undesired and detrimental to the economic system. Consequently, it is imperative to implement measures to preclude the system from transitioning into a chaotic state.

Various techniques have been employed to mitigate chaos within the supply chain, including the straight-line stabilization approach [53], the time-delay feedback control [49]. In this study, we adopt the time-delay feedback control method to regulate the chaos in system (16). The underlying principle of this strategy involves delaying a portion of the system's output signal and utilizing it as a feedback control signal for system regulation [47].

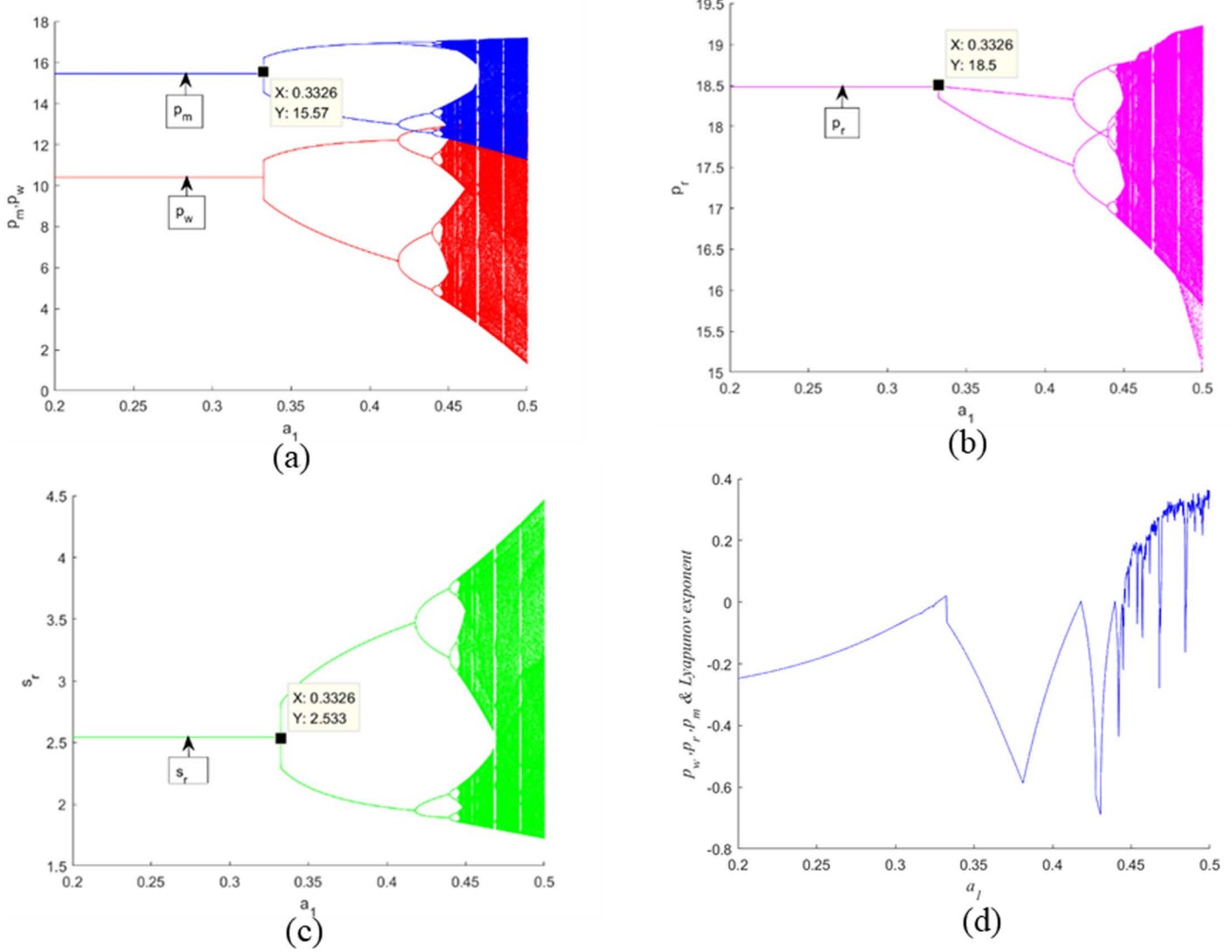

**Fig 4. Bifurcation diagrams and the correspo nding *LLE* of the dynamic system (16) for $\alpha_1$ when $\alpha_2 = 0.2$.**

To achieve this, we adjust the initial equation of system (16) by incorporating the system state variable feedback and parameter adjustment control approach, where $k_m > 0$ represents the control coefficient. Subsequently, the controlled version of system (17) can be expressed as follows:

$$
\begin{cases}
p_w(t+1) = (1-k_m)(p_w(t) + \alpha_1 p_w(t)\frac{\partial \pi_m(t)}{\partial p_w(t)}) + k_m \; p_w(t) \\
p_m(t+1) = (1-k_m)(p_m(t) + \alpha_2 p_m(t)\frac{\partial \pi_m(t)}{\partial p_m(t)}) + k_m \; p_m(t) \\
p_r(t) = \frac{-p_w(t)\gamma^2 + c_r\lambda\tau\gamma + \alpha\rho - \alpha\rho\theta + \mu\rho p_m(t) + \omega\rho p_w(t)}{2\omega\rho - \gamma^2} \\
s_r(t) = \frac{\alpha\gamma - \alpha\gamma\theta + \gamma\mu p_m(t) - \gamma\omega p_w(t) + 2c_r\lambda\omega\tau}{2\omega\rho - \gamma^2}
\end{cases}
\tag{17}
$$

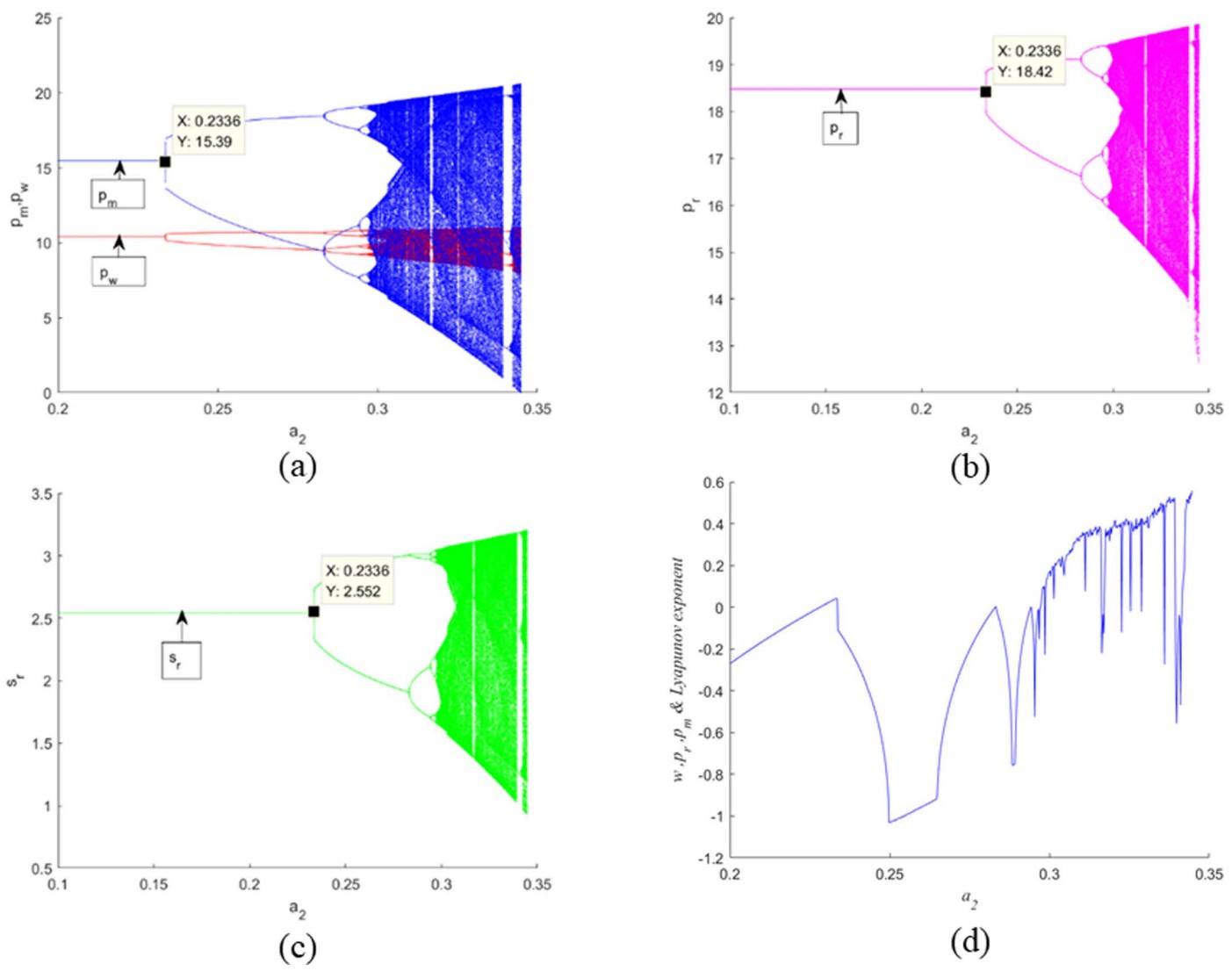

**Fig 5. Bifurcation diagrams and the corresponding _LLE_ of the dynamic system (16) with respect to its decision variables for $\alpha_2$ when $\alpha_1 = 0.17$ Fig 6 (a) and Fig 7 (a) represent the bifurcation diagrams of the profits of each member, where $\pi_m$ denotes the manufacturer's profit and $\pi_r$ denotes the retailer's profit.** Meanwhile, Fig 6(b) and Fig 7(b) depict the bifurcation diagrams of the total profit $\pi_t$. One sees that when $\alpha_1$ and $\alpha_2$ have small values, the profits gained by them are stable. With the increment of $\alpha_1$ and $\alpha_2$, the manufacturer's and retailer's performances worsen, and an inordinately rapid adjustment of $\alpha_1$ results in a steeper decline in their performances relative to $\alpha_2$. Furthermore, this heightens the unpredictability of the outcomes for all stakeholders, as evidenced in Figs 6 and 7, as well as increases the uncertainty results for all the players as displayed. Therefore, according to the Stackelberg model, it is advantageous for the manufacturer to keep the CLSC system in a stable condition in order to maximize expected profits and boost the efficiency of the CLSC system.

As illustrated in Fig 8(a) and 8(b), the period-halving bifurcation diagrams for channel prices and recycling service level, respectively, are depicted as functions of the control parameter $k_m$. It is evident that as $k_m$ increases, the chaotic behavior of the system is progressively mitigated. Specifically, when $k_m$ reaches 0.212, the system transitions from a chaotic state back to a stable state. This demonstrates that system state variable feedback and parameter adjustment control approach method is effective in eliminating chaos within the CLSC system. Furthermore, the corresponding _LLE_ becomes less than zero for $k_m$ values greater than 0.212, confirming that the system exits the chaotic regime and attains stability.

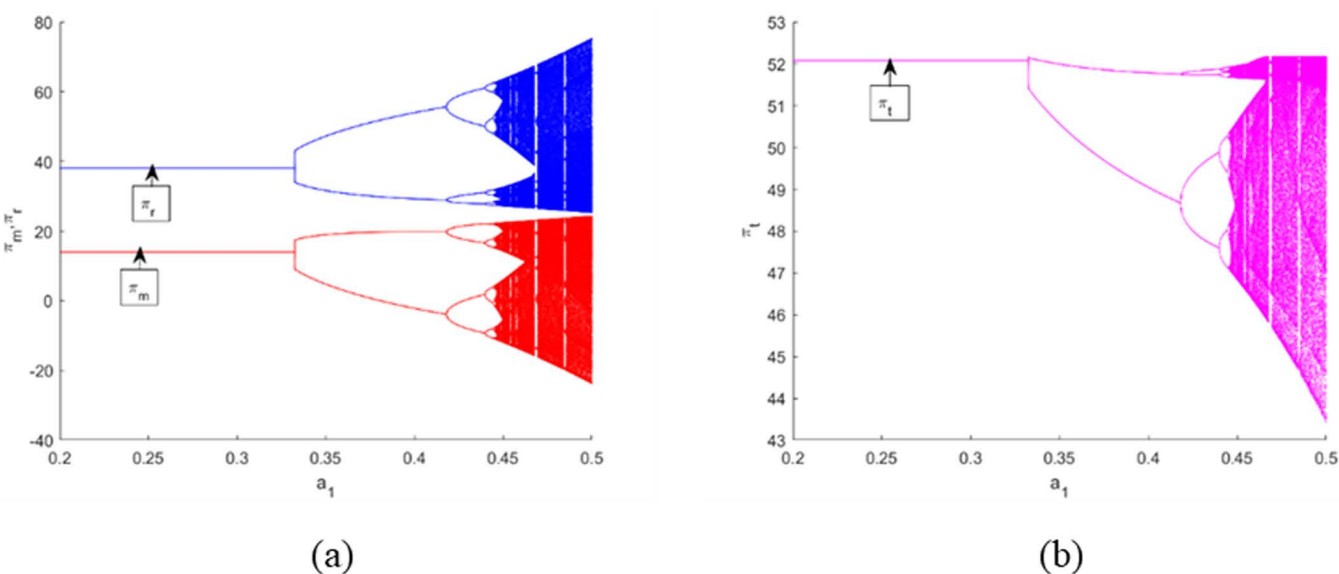

**Fig 6. Bifurcation diagrams and the corresponding _LLE_ of the dynamic system (16) with respect to the profits of each member and the supply chain for $\alpha_1$ when. $\alpha_2 = 0.2$.**

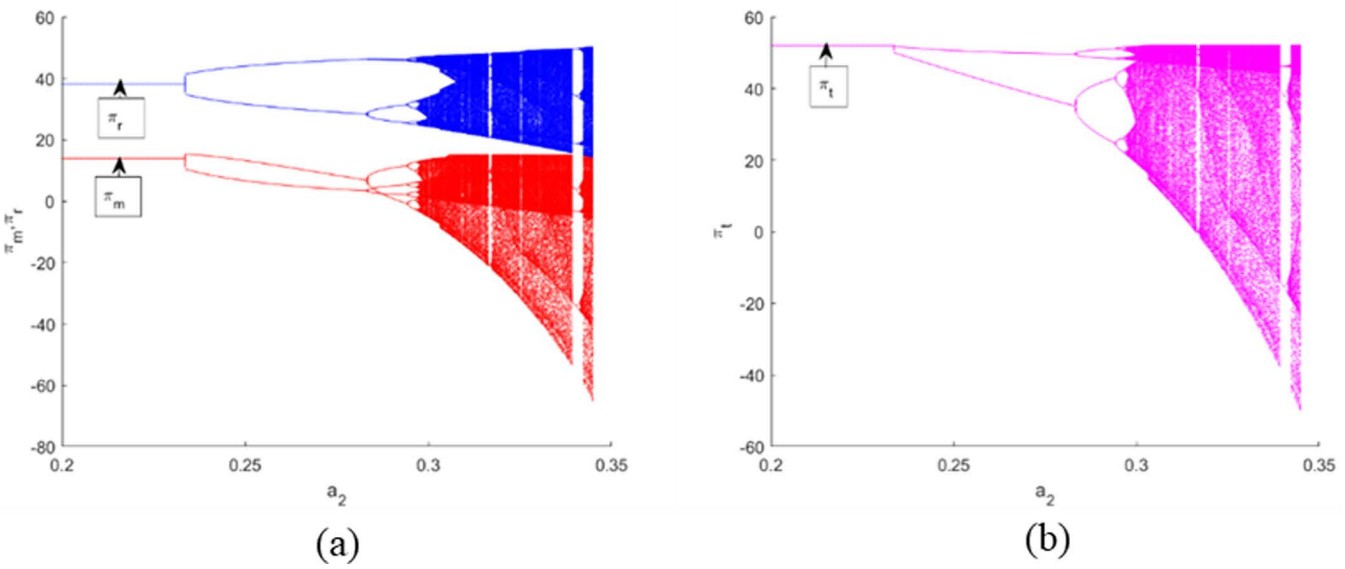

**Fig 7. Bifurcation diagrams and the corresponding _LLE_ of the dynamic system (16) with respect to the profits of each member and the supply chain for $\alpha_2$ when .$\alpha_1 = 0.17$.**

## 7. Conclusion

In the current paper, this paper study a static and dynamic Stackelberg game model where retailers provide recycling services within the "manufacturer-retailer" framework. Initially, we investigate the non-cooperative dynamic pricing strategies inherent in a CLSC comprising manufacturers and retailers. Subsequently, leveraging complex dynamic system theory, we conduct a comprehensive analysis of the impact of key parameters, including market potential size, the fund return rate,

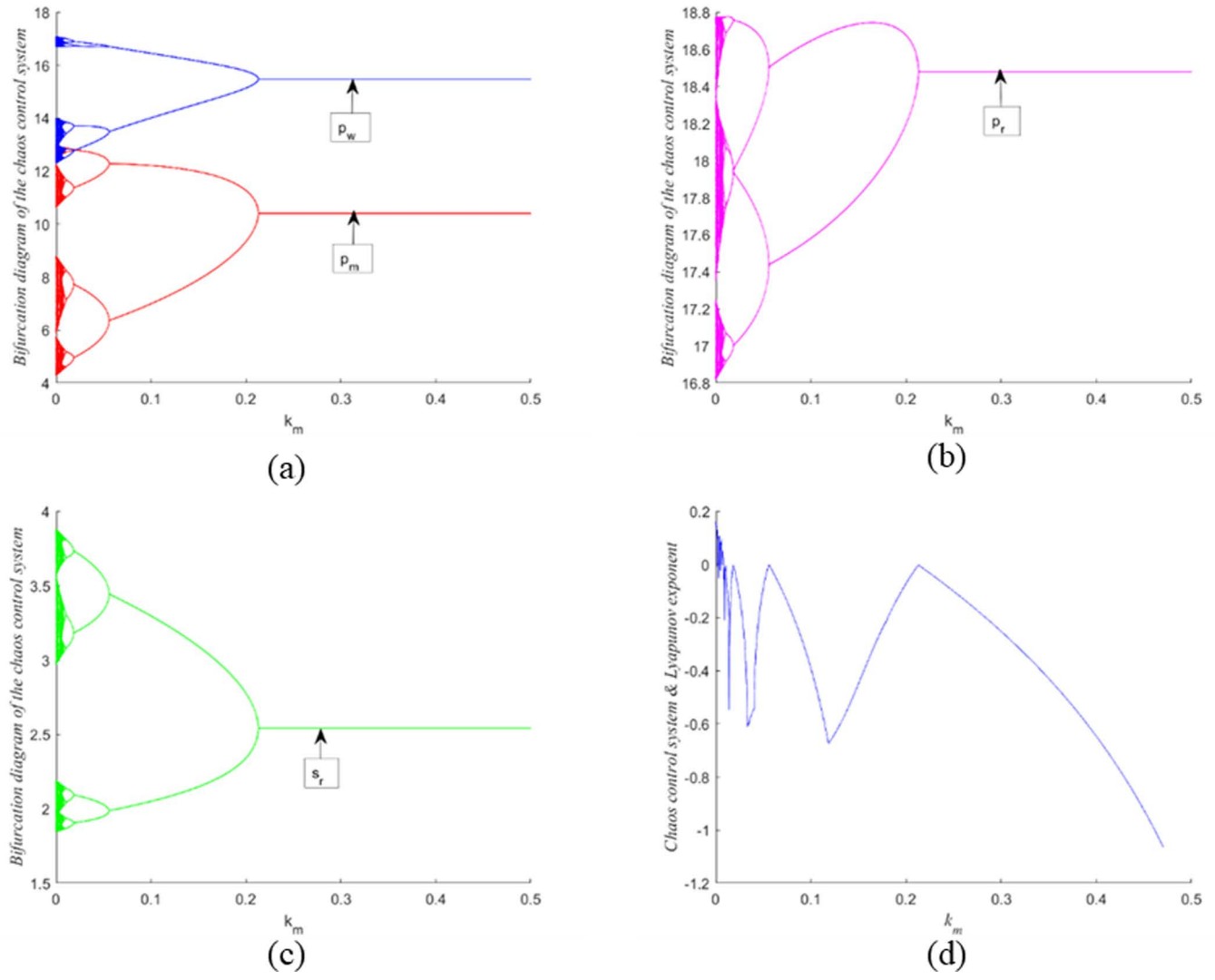

**Fig 8. Bifurcation diagram and the corresponding *LLE* with respect to the controlling coefficient $k_m$ when $\alpha_1 = 0.45$ and $\alpha_2 = 0.2$.**

and government subsidy criteria, on dynamic decision-making, along with an exploration of the inherent complexity. Furthermore, as various key parameters evolve, numerical simulations reveal that the dynamic game system exhibits characteristics such as bifurcation, chaos, and polymorphic stability. Our study has uncovered some intriguing findings, which are presented as follows:

(1) The conclusion drawn from this analysis is that, in the short term, government subsidy criteria act as a regulatory lever, decreasing direct sales channel prices $p_m$, while increasing retail channel prices $p_w$ and $p_r$, thereby promoting retail sales and suppressing direct sales price hikes, from a long-term perspective, excessive government subsidies can destabilize the system. Simultaneously, it positively impacts the level of electric Power battery recycling services offered by retailers due to reduced costs, increased returns, and enhanced competitiveness. Furthermore, the fund return rate s for manufacturers negatively affects their prices but positively influences retailers' prices, as retailers

offset costs and maintain competitiveness. Lastly, a larger potential market size leads to lower channel prices due to increased competition and cost savings from economies of scale.

(2) An excessively swift adjustment pace adopted by the manufacturer may propel the system beyond its stability threshold, resulting in chaotic outcomes. Additionally, the system exhibits greater sensitivity to the rate of direct sales price adjustments than to wholesale price. Given the manufacturer's role as the Stackelberg leader, their decision-making strategy exerts a profound impact on other members of the CLSC system. Therefore, Manufacturers, recognizing the CLSC system's higher sensitivity to direct sales price adjustments, should carefully calibrate these changes, prioritize pricing stability, emphasize value proposition, monitor market dynamics, consider long-term impacts, balance price and volume, enhance customer communication, and strengthen the maintenance of direct sales channels to navigate pricing complexities and ensure sustained success in the market.

(3) The impact of government subsidies on the stability region of the CLSC system is not significant. However, when government subsidies are excessively high, the stability region of the CLSC system will notably decrease, affecting the smooth operation of the supply chain. Therefore, the government should establish reasonable and scientific quotas for government subsidies.

(4) When the control parameter is appropriately selected, the approach involving the CLSC system state variable feedback and parameter adjustment can effectively eliminate the CLSC system chaos.

Based on the research conclusions presented above, the following recommendations are proposed:

(1) Impact of government subsidies and the fund return rate s for manufacturers:

At the government level, it is prudent to augment subsidies to manufacturers to an appropriate level, while exercising due diligence in policy formulation to prevent excessive allocations. This approach ensures that policies devised for retailers involved in recycling services do not compromise the stable functioning of the supply chain. In the short run, balanced subsidies can yield positive outcomes by lowering optimal prices, improving recycling services, and boosting the profits of supply chain participants. Conversely, imbalanced subsidies may have destabilizing consequences, leading to distortions in market prices, misallocation of resources, market disruptions, and fostering a long-term reliance on subsidies. Therefore, it is crucial for governments to adopt balanced intervention strategies to ensure market stability. And the government should closely monitor market dynamics and adjust subsidy amounts in a timely manner to avoid market disruptions caused by excessive subsidies. Alternatively, for manufacturers and retailers, manufacturers should consider the impact of the fund return rate s for manufacturers on pricing strategies, adjusting prices reasonably to balance profits and market share. Retailers, on the other hand, should leverage government subsidies to reduce costs, enhance competitiveness, and continuously improve the quality of recycling services.

(2) Relationship between manufacturer's pricing adjustment speed and system stability

For manufacturers, as Stackelberg leaders, they should fully recognize the impact of pricing adjustment speed on system stability and avoid chaotic outcomes resulting from overly rapid adjustments. Manufacturers should carefully calibrate price changes, prioritize pricing stability, formulate long-term strategies, and strengthen communication with retailers and consumers to navigate pricing complexities. Additionally, at the supply chain level, all members should enhance collaboration to jointly maintain system stability. This can be achieved through the establishment of information-sharing mechanisms, regular coordination meetings, and other means to facilitate communication and cooperation among members.

(3) Impact of government subsidies on the stable region of the CLSC systems

When formulating subsidy policies, the government should take into full consideration their impact on the stable region of the CLSC systems. A scientific subsidy quota system should be established to ensure that subsidies not only incentivize retailers to provide recycling services but also avoid causing excessive disruptions to system stability. Furthermore, the government and relevant agencies should periodically evaluate and adjust subsidy policies to ensure they adapt to market changes and effectively promote the sustainable development of the CLSC.

(4) Methods for Eliminating Chaos in the CLSC Systems:

From a technical and management perspective, it is crucial to strengthen the monitoring and feedback of the CLSC system state variables, promptly identifying and adjusting system parameters to eliminate chaos. Big data technologies, represented by blockchain technology, can be utilized to enhance the accuracy and efficiency of monitoring and adjustments. By leveraging the transparency, security, and traceability features of blockchain, as well as the analytical capabilities of big data, more precise and timely interventions can be made to maintain system stability. In addition,when formulating supply chain strategies, the possibility of chaos should be fully considered, and corresponding countermeasures should be developed. For example, an emergency response mechanism can be established to address the impact of unexpected situations on the supply chain, leveraging big data technologies to quickly assess and respond to disruptions.

This paper has made several significant contributions to the research on the CLSC System for Ev power battery recycling. It not only deepens the understanding of system dynamic decision-making, stability, and complexity but also proposes management strategies with practical guiding significance. However, there are certain constraints and avenues that merit further investigation. In our study, we only considered the scenario where the retailer provides recycling services; however, in actual operation, the retailer may offer such services concurrently. Additionally, illegal recycling platforms have not been taken into account in this paper. Besides, due to the swift progress in recovery, a number of third-party collectors have gained considerable influence to dominate the supply chain in certain sectors. Consequently, taking into account the varying channel power dynamics among decision-makers would yield greater insights for management. Additionally, in this paper, the secondary market is only treated as an exogenous variable; however, in reality, the secondary market is an important component of the CLSC system.

## Author contributions

**Data curation:** Xiaobin Wang, Huanying He, Lei Dai.

**Formal analysis:** Xiaobin Wang, Huanying He.

**Funding acquisition:** Xiaobin Wang.

**Investigation:** Xiaobin Wang.

**Methodology:** Xiaobin Wang.

**Project administration:** Xiaobin Wang.

**Resources:** Xiaobin Wang.

**Software:** Abudureheman Kadeer, Xiaobin Wang.

**Supervision:** Abudureheman Kadeer, Xiaobin Wang.

**Validation:** Abudureheman Kadeer, Xiaobin Wang.

**Visualization:** Abudureheman Kadeer, Xiaobin Wang.

**Writing – original draft:** Abudureheman Kadeer, Xiaobin Wang.

**Writing – review & editing:** Abudureheman Kadeer, Xiaobin Wang.

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
