## [Decision Letter · Decision Letter 0]

30 Oct 2024

PONE-D-24-39687Complexity Analysis of a Closed-loop Supply Chain for Power Battery Recycling under Government SubsidiesPLOS ONE

Dear Dr. wang,

Thank you for submitting your manuscript to PLOS ONE. After careful consideration, we feel that it has merit but does not fully meet PLOS ONE’s publication criteria as it currently stands. Therefore, we invite you to submit a revised version of the manuscript that addresses the points raised during the review process.

Please address all requested amendments by reviewers and revise your manuscript accordingly.Please submit your revised manuscript by Dec 14 2024 11:59PM. If you will need more time than this to complete your revisions, please reply to this message or contact the journal office at plosone@plos.org . Please include the following items when submitting your revised manuscript:

We look forward to receiving your revised manuscript.

Kind regards,

Dr Ashkan Memari

Academic Editor

PLOS ONE

Journal Requirements:

2. We note that your Data Availability Statement is currently as follows: 

All relevant data are within the manuscript and its Supporting Information files.

Reviewers' comments:

Reviewer's Responses to Questions

**Comments to the Author**

1. Is the manuscript technically sound, and do the data support the conclusions?

Reviewer #1: Yes

Reviewer #2: Partly

2. Has the statistical analysis been performed appropriately and rigorously? 

Reviewer #1: No

Reviewer #2: I Don't Know

3. Have the authors made all data underlying the findings in their manuscript fully available?

Reviewer #1: Yes

Reviewer #2: No

4. Is the manuscript presented in an intelligible fashion and written in standard English?

Reviewer #1: Yes

Reviewer #2: Yes

5. Review Comments to the Author

Reviewer #1: First and foremost, I would like to congratulate the work you have done, which contributes to the field you have worked in. Some improvements to be taken as mentioned below:

1. The abstract should quantify the results, and specify the scope and findings of the

sensitivity analysis.

2. The motivation and the justification for the research undertaken must be emphasized more in the introduction section. In addition, the paper's contribution must also be emphasized more, which should introduce the reader to the topic of the paper.

3. I suggest adding newer articles to the Literature Review section from 2024 to increase comprehensiveness.

4. Some main and sub-sections of the research were not numbered.

5. After the literature review section, research gaps and innovations should be mentioned.

6. Explain clearly the advantages and superiority of the proposed model.

7. The results of sensitivity analyses should be more explained.

8. The conclusion should reflect both the theoretical and practical significance of the manuscript.

9. Please revise your conclusion part into more details. Basically, you should enhance your contributions, limitations, underscore the scientific value added of your paper, and/or the applicability of your findings/results and future study in this session.

Reviewer #2: A/ The introduction is weak and does not adequately justify the need for the method. There is a lack of comprehensive background information on existing issues in resiliency and sustaibility, making it challenging for readers to appreciate the context and significance of the research. Merely referencing the Internet of Things does not inherently signify contributions.

B/ The literature section is not convincing in highlighting research gaps. Add and cite some refrences as below to enhance your related works, objectives and applications into your research:

- https://doi.org/10.1016/j.engappai.2024.108053

- https://doi.org/10.31181/jscda1120232

- https://doi.org/10.1016/j.asoc.2023.111012

C/ This research appears to focus primarily on the close-loop supply chain, with limited attention given to specific parameters related to the Static Stackelberg .Fore example I can't find any validation or justification how Static Stackelberg can assist to Supply Chain Efficiency??

D/ While defining the variables, do not use the same letter for different things with just changing the sets. Not a good practice.

E/ How did you solve this problem? Which solver? How much time does it take? Why did you not use the Mixed Integer Programming model?

6. PLOS authors have the option to publish the peer review history of their article (what does this mean? ). If published, this will include your full peer review and any attached files.

**Do you want your identity to be public for this peer review?** For information about this choice, including consent withdrawal, please see our Privacy Policy .

Reviewer #1: No

Reviewer #2: No

---

## [Author Response · Author response to Decision Letter 1]

8 Jan 2025

We appreciate the constructive feedback provided by the reviewers. Below, we address each of the points raised and outline the improvements we plan to make to the manuscript.

Reviewer #1:

1.Abstract Quantification and Scope Clarification:

Response: We have revised the abstract to include a mention of the sensitivity analysis, emphasizing that although specific quantitative conclusions are not presented, targeted viewpoints based on the analysis have been put forth, and the scope of the research has been clearly delineated.

2.Enhanced Motivation and Justification:

Response: We have strengthened the introduction section by emphasizing the motivation and justification for the research. We have clearly articulated the research problem, its significance, and how our work contributes to the field.

3.Updated Literature Review:

Response: We have incorporated newer articles from 2024 into the literature review section to ensure comprehensiveness and up-to-date coverage of the topic.

4.Section Numbering:

Response: We have added numbering to all main and sub-sections to improve the organization and readability of the manuscript.

5.Research Gaps and Innovations:

Response: We have explicitly mentioned the research gaps and innovations that our work addresses following the literature review section.

6.Model Advantages and Superiority:

Response: We have clearly explained the advantages and superiority of the proposed model, highlighting its unique features and benefits over existing approaches.

7.Detailed Sensitivity Analysis Results:

Response: We have provided a more detailed explanation of the results from the sensitivity analyses, including an analysis of the sensitivity of profits among members, discussing their implications and how they inform our understanding of the model's behavior in a more practical context.

8.Theoretical and Practical Significance in Conclusion:

Response: We have revised the conclusion to reflect both the theoretical and practical significance of the manuscript, emphasizing the contributions and implications of our findings.

9.Detailed Conclusion:

Response: We have enhanced the conclusion section by providing more details on our contributions, limitations, the scientific value added by our paper, the applicability of our findings, and suggestions for future research.

Reviewer #2:

A. Strengthening the Introduction:

Response: We have enhanced the introduction by providing a more comprehensive background on existing issues in resiliency and sustainability. We have clearly justified the need for our method.

B. Enhanced Literature Section:

Response: We have incorporated the suggested references, along with other recent literature, into the literature section to better highlight research gaps, objectives, and applications of our research.

C. Focus on Static Stackelberg and Supply Chain Efficiency:

Response: In response to your concern, we have already conducted numerical simulations to validate three key parameters: government subsidy standards, fund return rates, and market size, primarily utilizing mathematical monotonicity (where the partial derivative is greater than 0). Additionally, we have performed sensitivity analyses on members' profits, leading to a key conclusion: In the short term, policy subsidies will increase supply chain members' profits and lower prices, but in the long term, excessive government subsidies can cause the system to enter a state of chaos, harming the profits of all parties and the overall system. Regarding your point about limited attention to specific key parameters, we acknowledge the validity of your observation. In fact, I am currently working on another article where I plan to enhance the focus on these key parameters.

D. Variable Definition Clarity:

Response: We will revise the variable definitions to ensure that different concepts are not represented by the same letter with only set changes. We will use distinct notation to avoid any confusion.

E. Problem Solving Approach and Solver Details:

Response: After extensively reviewing the literature and consulting specialized textbooks on mixed-integer programming (MIP), I acknowledge that MIP models have demonstrated effective applications in solving location and pricing problems in supply chain management. However, in the context of our current research, I believe that adopting an MIP model is not the optimal choice. Below are the specific reasons why I do not advocate for the use of an MIP model:

(1) Mismatch in Problem Nature

MIP models primarily address optimization problems, aiming to maximize or minimize a linear function subject to a series of linear constraints, with the assumption that parameters such as demand and cost are deterministic. Nevertheless, in our study, we are more concerned with the strategic interactions and decision-making sequences among multiple participants, where the key parameters are uncertain and the relationships between variables exhibit significant nonlinearity. The Stackelberg game model is particularly adept at capturing such strategic interactions and decision-making sequences, making it more suitable for analyzing our current problem.

(2) Characteristics of Decision Variables

While MIP models can accommodate both continuous and discrete (integer) decision variables, the focus of our current problem lies in the strategic choices made by participants, which often involve complex decision-making logic and dynamic changes, rather than simple quantities or counts. By utilizing nonlinear system theory, the Stackelberg game model can better handle these complex decision variables and account for the mutual influences among participants.

(3) Limitations of Solution Methods

Common solution methods for MIP models, including branch-and-bound, cutting-plane, and interior-point methods, excel at searching the feasible solution space and finding optimal or near-optimal solutions. However, in our current problem, we are interested in analyzing the strategic interactions and decision-making processes among participants, rather than merely finding an optimal solution. Furthermore, we need to analyze the stability of fixed points and the evolution of equilibrium strategies. The solution methods for the Stackelberg game model, such as backward induction, provide deeper insights into the decision-making logic and strategic choices among participants, making them more suitable for solving our current problem.

(4) Differences in Application Domains

MIP models are more focused on optimization and resource allocation problems, such as production planning, logistics planning, and resource allocation. In contrast, our current problem involves strategy formulation and decision analysis in areas such as market competition, traffic flow control, and network security. In these domains, the Stackelberg game model can better capture the strategic interactions and decision-making sequences among participants, providing more powerful strategic advice to decision-makers.

In summary, I believe that adopting an MIP model is not the best choice for our current project or problem. Instead, the Stackelberg game model can better meet our needs by capturing the strategic interactions, decision-making sequences, and complex behaviors of system evolution among participants, providing more accurate strategic advice to decision-makers. Therefore, I recommend using the Stackelberg game model for analysis and solution.

We thank the reviewers for their valuable feedback, which will significantly improve the quality and clarity of our manuscript. We will make the necessary revisions and resubmit the paper for further consideration.

---

## [Decision Letter · Decision Letter 1]

30 Mar 2025

Complexity Analysis of a Closed-loop Supply Chain for Power Battery Recycling under Government Subsidies

PONE-D-24-39687R1

Dear Dr. Wang,

We’re pleased to inform you that your manuscript has been judged scientifically suitable for publication and will be formally accepted for publication once it meets all outstanding technical requirements.

Kind regards,

João Zambujal-Oliveira

Academic Editor

PLOS ONE

Additional Editor Comments (optional):

Reviewers' comments:

Reviewer's Responses to Questions

**Comments to the Author**

1. If the authors have adequately addressed your comments raised in a previous round of review and you feel that this manuscript is now acceptable for publication, you may indicate that here to bypass the “Comments to the Author” section, enter your conflict of interest statement in the “Confidential to Editor” section, and submit your "Accept" recommendation.

Reviewer #1: (No Response)

Reviewer #2: All comments have been addressed

2. Is the manuscript technically sound, and do the data support the conclusions?

Reviewer #1: Yes

Reviewer #2: Yes

3. Has the statistical analysis been performed appropriately and rigorously? 

Reviewer #1: Yes

Reviewer #2: Yes

4. Have the authors made all data underlying the findings in their manuscript fully available?

Reviewer #1: (No Response)

Reviewer #2: Yes

5. Is the manuscript presented in an intelligible fashion and written in standard English?

Reviewer #1: Yes

Reviewer #2: Yes

6. Review Comments to the Author

Reviewer #1: Dear Authors,

I appreciate the thorough revisions you have made in response to the feedback. The updated manuscript demonstrates a clear improvement in terms of argumentation, methodology, and overall presentation. Your responses effectively addressed the concerns raised, and I believe the paper now meets the required standards for publication.

Best wishes

Reviewer #2: The authors have addressed my concerns, and I fully support this version of the paper for publication.

7. PLOS authors have the option to publish the peer review history of their article (what does this mean? ). If published, this will include your full peer review and any attached files.

**Do you want your identity to be public for this peer review?** For information about this choice, including consent withdrawal, please see our Privacy Policy .

Reviewer #1: No

Reviewer #2: No

---

## [Editor Report · Acceptance letter]

PONE-D-24-39687R1

PLOS ONE

Dear Dr. Wang,

I'm pleased to inform you that your manuscript has been deemed suitable for publication in PLOS ONE. Congratulations! Your manuscript is now being handed over to our production team.

Kind regards,

on behalf of

Prof. João Zambujal-Oliveira

Academic Editor

PLOS ONE